# Modeling unveils sex differences of signaling networks in mouse embryonic stem cells

Zeba Sultana[1], Mathurin Dorel[2], Bertram Klinger[2] (ID), Anja Sieber[2] (ID), Ilona Dunkel[1], Nils Blüthgen[2,*] (ID) & Edda G Schulz[1,**] (ID)

## Abstract

**For a short period during early development of mammalian embryos, both X chromosomes in females are active, before dosage compensation is ensured through X-chromosome inactivation. In female mouse embryonic stem cells (mESCs), which carry two active X chromosomes, increased X-dosage affects cell signaling and impairs differentiation. The underlying mechanisms, however, remain poorly understood. To dissect X-dosage effects on the signaling network in mESCs, we combine systematic perturbation experiments with mathematical modeling. We quantify the response to a variety of inhibitors and growth factors for cells with one (XO) or two X chromosomes (XX). We then build models of the signaling networks in XX and XO cells through a semi-quantitative modeling approach based on modular response analysis. We identify a novel negative feedback in the PI3K/AKT pathway through GSK3. Moreover, the presence of a single active X makes mESCs more sensitive to the differentiation-promoting Activin A signal and leads to a stronger RAF1-mediated negative feedback in the FGF-triggered MAPK pathway. The differential response to these differentiation-promoting pathways can explain the impaired differentiation propensity of female mESCs.**

**Keywords** cell signaling; mathematical modeling; pluripotency; stem cells; X-chromosome inactivation; X chromosome

**Subject Categories** Signal Transduction; Stem Cells & Regenerative Medicine

**Mol Syst Biol. (2023) 19: e11510**

## Introduction

In mammals such as humans and mice, sex is determined by a pair of heteromorphic sex chromosomes X and Y, which are highly divergent in their gene content. As a result, males carrying an XY sex-chromosome composition have only a single copy of X-linked genes, while XX females have two. To ensure equivalent levels of gene expression from the X chromosome in both sexes, one of the two X chromosomes in female cells is transcriptionally silenced at the early stages of embryonic development—a phenomenon known as X-chromosome inactivation (Schulz & Heard, 2013). While X-dosage differences are thus largely compensated in somatic cells, X-dosage-effects are thought to contribute to sex differences in early development, prior to X inactivation (Schulz *et al*, 2014; Schulz, 2017; Song *et al*, 2019). Since double X-dosage interferes with cell differentiation (Schulz *et al*, 2014), early development of embryos with two X chromosomes is slightly delayed compared to males or XO embryos, which contain only one X chromosome (Burgoyne *et al*, 1995).

In mice, both X chromosomes are active in pluripotent cells at the late blastocyst stage of female embryos (Rastan, 1982; Mak *et al*, 2004; Kobayashi *et al*, 2016). This stage can be modeled in cell culture using murine embryonic stem cells (mESCs), where both X chromosomes are active in female lines (Boroviak & Nichols, 2014; Ying & Smith, 2017). mESCs are characterized by their ability for long-term self-renewal, i.e. an unlimited number of symmetrical cell divisions without differentiating, and pluripotency, allowing them to give rise to differentiated cell types of all three germ layers. The pluripotent state is maintained through pluripotency factors, such as NANOG, OCT4, and SOX2 (Avilion *et al*, 2003; Loh *et al*, 2006; Silva *et al*, 2009; Yeo & Ng, 2013), but their sustained expression requires specific culture conditions. Conventionally, mESCs are cultured in the presence of leukemia inhibitory factors (LIF), which signals through the JAK/STAT3 pathway (Smith *et al*, 1988; Martello *et al*, 2013). A more homogeneous population of cells in a so-called "naive" pluripotent state can be maintained through addition of two inhibitors (2i) that block MEK, a central component of the MAPK signaling pathway, and GSK3, which is regulated by WNT and the AKT/mTOR pathway (Ying *et al*, 2008; Marks *et al*, 2012). The naive state is characterized by homogenous high expression of a subset of pluripotency factors, including NANOG, and by strongly reduced global DNA methylation levels (Ying *et al*, 2008; Ficz *et al*, 2013; Leitch *et al*, 2013).

Female mESCs adopt a more naive state compared to males, even when cultured in conventional ES medium, due to the presence of two active X chromosomes (Schulz *et al*, 2014). This is evident by

---

1 Systems Epigenetics, Otto-Warburg-Laboratories, Max Planck Institute for Molecular Genetics, Berlin, Germany
2 Computational Modelling in Medicine, Institute of Pathology, Charité-Universitätsmedizin Berlin, Berlin, Germany
 *Corresponding author. Tel: +49-30-2093-92390; E-mail: nils.bluethgen@charite.de
 **Corresponding author. Tel: +49-30-8413-1226; E-mail: edda.schulz@molgen.mpg.de

higher expression of pluripotency factors, lower DNA methylation levels, and differential activity of several signaling pathways (Zvetkova *et al*, 2005; Schulz *et al*, 2014). Specifically, cells with two active X chromosomes show increased activity of the pluripotency-associated AKT pathway and reduced activity of the differentiation-promoting MAPK pathway. Recently, we and others have identified X-linked genes that contribute to these X-dosage effects, which code for DUSP9, a negative regulator of the MAPK pathway, and KLHL13, an E3 ubiquitin ligase adaptor protein (Choi *et al*, 2017; Genolet *et al*, 2021). Overall, the presence of two active X chromosomes in female mESCs confers a more naive pluripotent state and delays their exit from pluripotency (Schulz *et al*, 2014).

To understand how the different states of the signaling network in cells with one and two X chromosomes arise, we aimed to reconstruct these networks in cells carrying one or two copies of the X chromosome and identify links with differential activity using mathematical modeling. We used modular response analysis (MRA; Kholodenko *et al*, 1997; Bruggeman *et al*, 2002), which is a modeling framework that lies in between the qualitative nature of Boolean models and detailed mechanistic approaches, such as ordinary differential equation (ODE) models. Using the statistical framework of maximum likelihood and likelihood ratio tests enabled the direct quantitative comparison of signaling networks of two mESC lines that differ in their X-chromosomal dosage (Stelniec-Klotz *et al*, 2012; Klinger *et al*, 2013; Dorel *et al*, 2018). Modeling predicted the presence of additional links in the literature-derived network including a novel feedback loop from GSK3 to the IGFR pathway upstream of AKT. Furthermore, we found that cells with a single active X chromosome show a stronger response to Activin and that the feedback loop in the MAPK pathway is stronger in such cells, explaining the previously reported delayed exit from pluripotency in the case of female cells where X inactivation has not yet been accomplished.

# Results

## Systematic perturbation of the signaling network in mESCs

To unravel the reasons for sex differences in mESCs (Schulz *et al*, 2014), we set out to understand how X-chromosomal dosage modulates the signaling network in this cell type. To this end, we aimed at building mathematical models of the mESC signaling network in cells with one (XO) and two (XX) X chromosomes. We employed MRA, which provides an approach to reverse-engineer networks based on systematic perturbation data (Kholodenko *et al*, 1997; Bruggeman *et al*, 2002; Santra *et al*, 2018) and which has been successfully applied to signaling pathways (Klinger *et al*, 2013; Dorel *et al*, 2018, 2021; Brandt *et al*, 2019; Hood *et al*, 2019; Berlak *et al*, 2022). We included five signal transduction pathways in the analysis that have been shown to control pluripotency or differentiation in mESCs (Fig 1A). These include three pathways that stabilize the pluripotent state, namely LIF-mediated activation of JAK/STAT3 (Niwa *et al*, 2009; Martello *et al*, 2013), IGF1-mediated activation of PI3K/AKT that results in phosphorylation of mTOR and GSK3 (Paling *et al*, 2004; Niwa *et al*, 2009) and BMP4-mediated phosphorylation of SMAD1/5/9 (Ying *et al*, 2003; Li *et al*, 2012). In addition, we included two pathways that drive differentiation, namely FGF4-mediated activation of the MEK/ERK

cascade (Kunath *et al*, 2007; Kim *et al*, 2012; Hamilton *et al*, 2013) and Activin A (ActA)-mediated phosphorylation of SMAD2/3 (Gadue *et al*, 2006; Kunath *et al*, 2007; Fei *et al*, 2010). Although the WNT pathway also plays an important role during early development, it was not included in our analysis, because it signals via protein stabilization (Sato *et al*, 2004). It therefore operates at a slower time scale than signals transmitted via post-translational modifications and could not be interrogated with the short perturbations we used (see below).

To generate the systematic perturbation data required for MRA-based network reconstruction, we aimed at identifying at least one perturbation (growth factor stimulation or enzyme inhibition) and one robust activity readout (phospho-protein) per pathway. We therefore tested ligands that stimulate these pathways and several inhibitors of pathway components (Fig EV1, Table EV1). We treated the female mESC line 1.8 XX with different doses of each ligand or inhibitor and measured the effect on phosphorylation levels of a downstream pathway component.

We then selected a subset of treatments (JAKi, IGFRi, Pi3Ki, FGFRi, MEKi, BMP4Ri, FGF4, ActA) for which we observed a clear response (Fig EV1A–H). As increasing concentrations of IGF1 and BMP4 did not show a corresponding increase in the phosphorylation of AKT and SMAD1/5/9, respectively (Fig EV1I and J), these ligands were not used for the systematic perturbation experiment. Since mESC media is supplemented with LIF, one of the five signaling pathways included in our network is constantly stimulated. We therefore decided to perturb this pathway by withdrawal of the cytokine (referred to as NoLIF).

We treated mESCs with one (1.8 XO) and two X chromosomes (1.8 XX) with either a ligand or an inhibitor or a combination of two treatments (except IGFRi/PI3Ki and FGFRi/MEKi, which inhibit the same pathway), resulting in 53 treatments per cell line (Fig 1B). The duration of the treatment was selected to be 30 min to allow the system, which relies on fast post-translational changes, to attain a new approximate steady state, while preventing extensive transcriptional and translational effects.

To assess how each perturbation affected the signaling network, the resulting changes in phosphorylation levels were quantified for seven pathway components. AKT, GSK3, mTOR, and MEK phosphorylation was assayed with a Luminex proteomics platform using a bead-based multiplex assay that allows simultaneous measurement of multiple phosphoproteins in a sample lysate. ERK, STAT3, and SMAD2 phosphorylation was measured by two-color Western blotting using infra-red-dye-labeled secondary antibodies. In this way, we could assess the activity of all selected pathways with the exception of BMP4 signaling, where the pSMAD1/5/9 signal was too weak for robust quantification (Figs 1A, and EV1H and J). With 53 perturbations and seven readouts, we thus collected 371 data points for each cell line in 3 biological replicates. For each data point, we calculated the mean fold change relative to the DMSO control (Datasets EV1 and EV2).

To assess the quality of the collected data, we analyzed whether the single treatments altered activity of known downstream targets. We indeed observed most of the expected effects (Fig 1C, dashed boxes), including reduced AKT, GSK3, and mTOR phosphorylation in response to IGFR and PI3K inhibition, reduced MEK and ERK phosphorylation upon FGFR inhibition, reduced STAT3 phosphorylation upon JAK inhibition or LIF withdrawal, and increased SMAD2

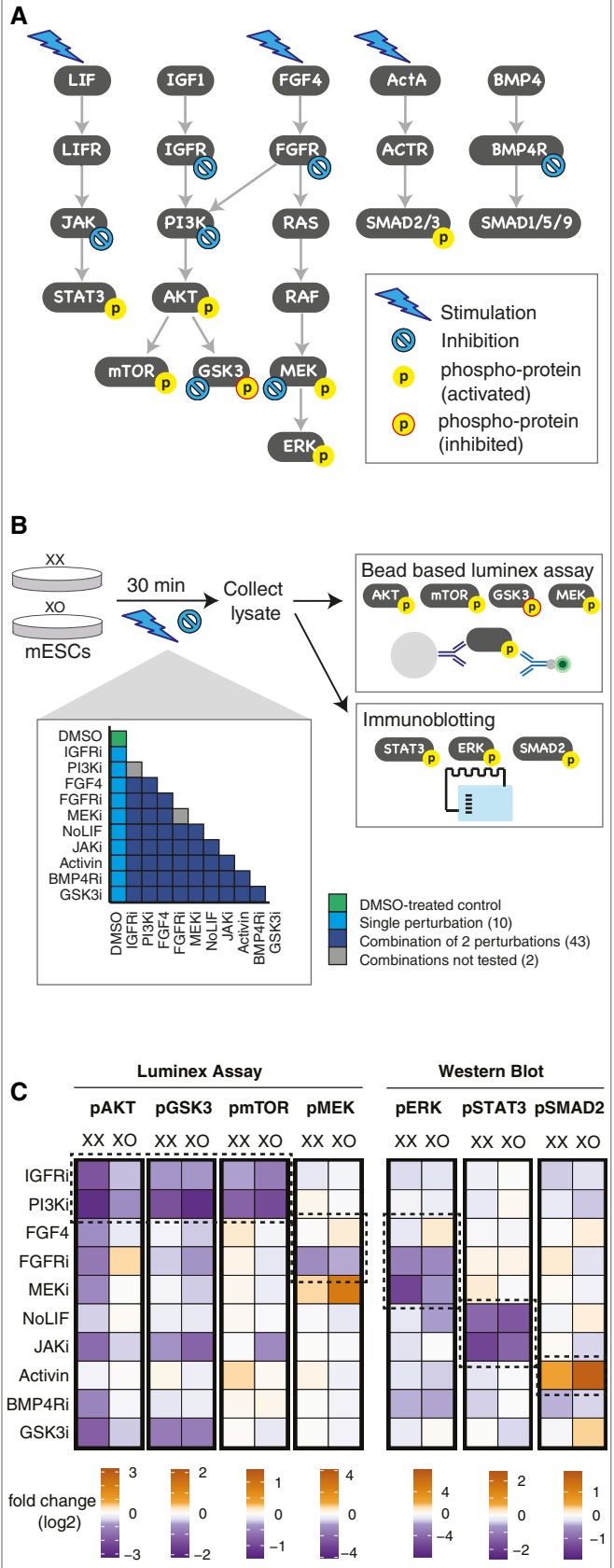

**Figure 1. Generation of systematic perturbation data.**

A Signaling pathways known to play a role in the maintenance of pluripotency or initiation of differentiation in mESCs. Perturbations (stimulation, inhibition) and activity readouts (phospho-sites) are indicated.

B Overview of the systematic perturbation experiment. A female mESC line (1.8XX) and its subclone carrying a single copy of the X chromosome (1.8XO) cultured in LIF-containing media were treated with either DMSO, one of the 10 single perturbations or a combination of two perturbations (all combinations except PI3Ki + IGFRi and FGFRi + MEKi were used as indicated). After 30 min treatment, cell lysates were collected and protein phosphorylation was measured using either a bead-based Luminex assay (light gray bead; green fluorophore; blue antibodies) or by immunoblotting. Inhibitors/ligands and their concentrations used: PI3K inhibitor LY294002 (5 µM), IGFR inhibitor OSI-906/Linsitinib (10 µM), FGF receptor inhibitor CH5183284/Debio-1347 (0.2 µM), MEK inhibitor U0126 (5 µM), JAK Inhibitor I (1 µM), BMP4 receptor inhibitor LDN-193189 (1 µM), GSK3 inhibitor, CHIR99021/CT99021 (6 µM), FGF4 (10 ng/ml), Activin (15 ng/ml). The experiment was performed in three independent replicates.

C The mean fold change across three biological replicates in phosphorylation in response to single perturbations in the two cell lines relative to their respective DMSO-treated controls. Responses expected based on the network in panel (A) are highlighted (dashed boxes).

Source data are available online for this figure.

phosphorylation upon ActA treatment. Only FGF4 treatment did not trigger the expected increase in pMEK and pERK levels in XX cells, maybe due to the higher basal phosphorylation levels of these pathway intermediates in the XX context (Schulz *et al*, 2014). Nevertheless, we concluded that the data set generally showed the expected trends. Additionally, we observed feedback regulation and crosstalk between pathways, such as pMEK increase upon MEKi treatment, as previously reported (Fritsche-Guenther *et al*, 2011; Lito *et al*, 2014), a reduction of pGSK3 and pAKT upon GSK3i treatment and a decrease in AKT, GSK3 and mTOR phosphorylation upon JAK inhibition (Fig 1C). In summary, we generated a large systematic perturbation data set in two mESC lines carrying one and two X chromosomes, respectively, as the basis for network reconstruction.

**Reconstruction of the signaling network in XX and XO mESCs**

In the next step, we developed a model of the signaling network in mESCs. We started from a literature-based network, which was then extended in a way to best reproduce the perturbation data (Fig 2A). The starting network comprised the five signaling pathways that were covered by the perturbation data set. We mostly included the well-established canonical linear signaling cascades, but little crosstalk and no feedback loops (Fig 1A). This allowed us to test whether our network extension procedure could rediscover known links, such as crosstalk from LIF towards MEK/ERK and AKT activation (Niwa *et al*, 2009) or feedback inhibition within the MEK/ERK pathway (Dougherty *et al*, 2005; Fritsche-Guenther *et al*, 2011; Lito *et al*, 2014; Schulz *et al*, 2014; Nett *et al*, 2018).

First, we estimated the model parameters for the literature-derived network structure for XX and XO cells separately (Fig 2A, Network parameterization). Shortly, we estimated local response coefficients, parameters describing the direct linear effects of one node on another, by minimizing the error-standardized discrepancy between the model values and the measurements (see Materials and

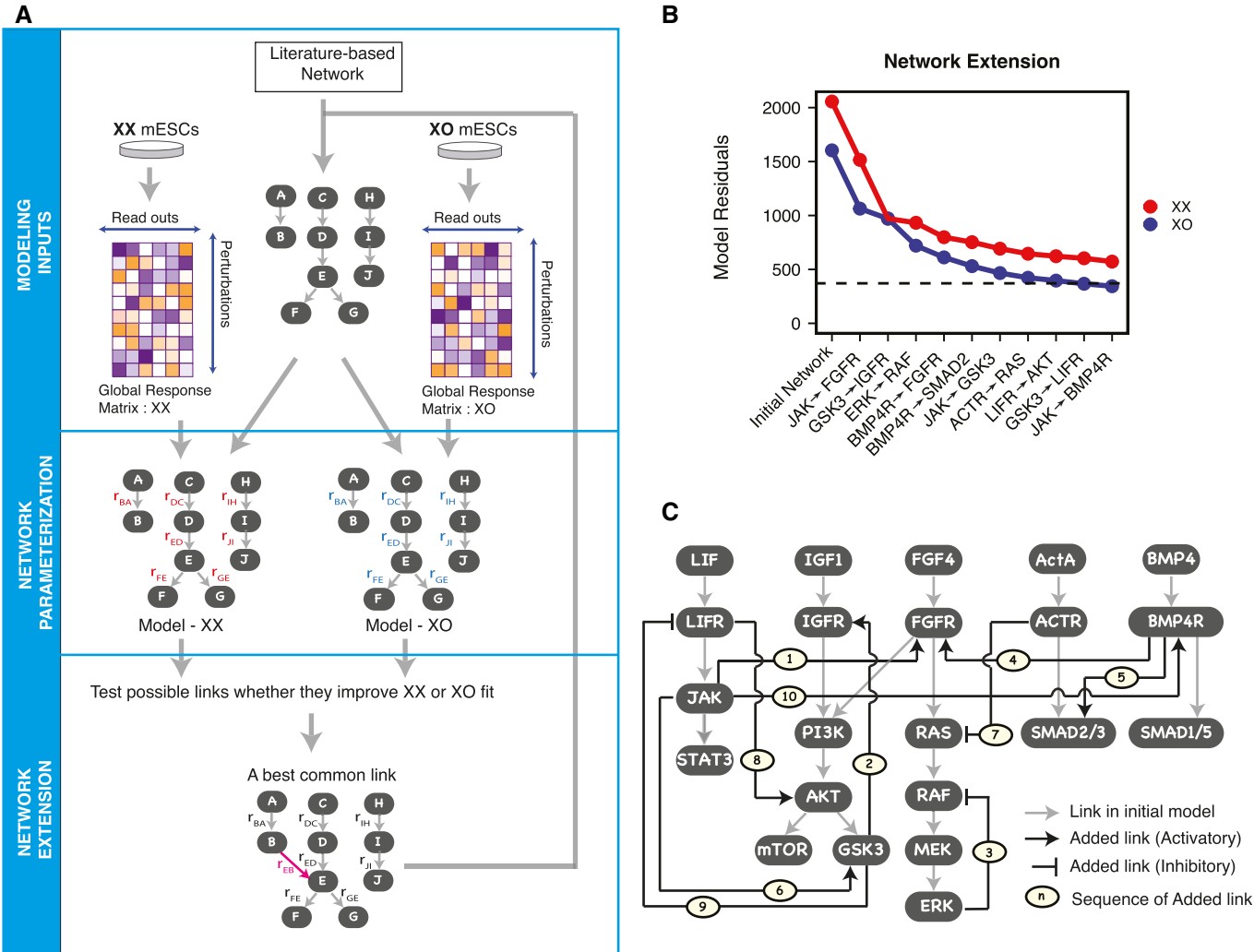

**Figure 2. Identification of novel links within the mESCs signaling network.**

A   Flow chart describing the steps followed for modeling the signaling networks in XX and XO cells: Input for each cell line was its perturbation data set (Global response matrix) and the literature-based network. Two cell line-specific models were parameterized using STASNet (Network Parameterization). Additional links were explored to find those that improved fit to the experimental data. The link that most improved the fit of the XX and the XO model was added to the network (Network extension). The steps were repeated until no such common links could be found.

B   Through the procedure shown in (A), 10 links were added to the literature-based network (x-axis). Change in model residuals with each link addition is plotted (y-axis). The dotted line represents the expected mean residuals, if the model explains all the data (equal to the number of data points minus the number of parameters).

C   The completed network structure, found with the procedure outlined in (A). Links shown in gray are already present in the literature network in Fig 1A. Newly added links (black) are numbered in the sequence in which they were added.

Methods for details). We used the R-package STeady-STate Analysis of Signaling Networks (STASNet), which is a maximum likelihood implementation of MRA (Dorel *et al*, 2018). In multiple rounds of network completion, we then added additional links to the starting network that reduced the model residuals. Throughout the optimization process, we assumed that the structure of the network would be identical in isogenic XX and XO cell lines, and differences in response to perturbations would arise from differences in model parameters. In each iteration, we therefore tested all possible links for the XX and XO models and then added the link that led to the strongest decrease in the summed residuals of both models (Fig 2A, Network extension). This process was repeated until we could not

find any common links that resulted in a statistically significant decrease of model residuals in both cell lines (likelihood ratio test, Benjamini–Hochberg adjusted *P* < 0.005).

A total of 10 additional links were added to the network, resulting In 72 and 78 % reduction of the residuals of the XX and XO models, respectively (Fig 2B and C). These included links that had been reported previously in mESCs, such as crosstalk from the LIF pathway towards MAPK signaling (Link 1: JAK → FGFR) and AKT signaling (Link 8: LIFR → AKT) (Niwa *et al*, 2009) and a feedback loop in the MAPK pathway (Link 3: ERK⁻|RAF; Dougherty *et al*, 2005; Fritsche-Guenther *et al*, 2011; Lito *et al*, 2014; Schulz *et al*, 2014; Nett *et al*, 2018). Apart from these links, which had been

previously reported in mESCs, the network extension procedure also identified novel interactions such as a putative feedback loop in the AKT pathway (Link 2: GSK3 → IGFR) and crosstalk from the Activin receptor towards MAPK signaling (Link 7: ACTR ⊣ RAS) among others.

Comparison between the experimental data and model simulations revealed that the completed network with 10 additional links could recapitulate many aspects of the experimental observations better than the initial literature-derived model (Fig 3A). For instance, the GSK3 inhibitor-induced decrease in phosphorylation of AKT and GSK3 in XX cells is more comparable with the experimental data in the completed model than in the initial model (Fig 3A, pAKT/pGSK3 arrows). More noticeably, the increase in MEK phosphorylation by MEK inhibitor treatment in both cell lines and to a higher magnitude in XO cells is not seen in the initial model but recapitulated well in the completed model (Fig 3A, pMEK, arrows). Similarly, decrease in SMAD2 phosphorylation with BMP4 receptor inhibitor treatment was not seen in the initial models but is recapitulated in the completed models (Fig 3A, pSMAD2, arrows). For a more systematic comparison between the initial and completed models, we calculated the coefficient of determination, $R^2$, between data and simulation for each analyte (Fig 3B). For pAKT, pGSK3, pMEK, and pERK, the agreement between model and experimental measurements was strongly improved through the network completion (increase in $R^2$). For pmTOR, pSTAT3, and pSMAD2, the agreement was already good for both cell lines in the initial models and was further improved in their completed models, except for pmTOR in case of XX cells.

We also analyzed for each readout the treatment, for which data and initial model were most discordant (Fig 3C and D). Here the completed model could recapitulate the experimental measurements better than the initial model for 5 and 6 out of 7 readouts for XX and XO cells, respectively. Finally, we analyzed for each readout the treatment to which XX and XO cells responded most differently in the experimental data (Fig 3E). In the majority of cases, the completed models could capture the differential responses, except for pmTOR and pSTAT3. Overall, the completed model could reproduce most aspects of the experimental data, and the added links clearly improved model performance.

To ensure the robustness of the analysis, we repeated the procedure for network completion with different *P*-value thresholds used for filtering links to be added to the network (Fig EV2). The first six links added in our analysis were robustly identified with all threshold values, with exception of the feedback loop within the MAPK pathway, which was implemented in a slightly different manner in the different runs (originating from ERK or MEK and targeting RAF or MEK). The last four links by contrast were only found in a subset of runs. For the subsequent model analysis and model testing, we therefore focused on the first six links. Taken together, our results clearly show that the additional links our network reconstruction approach identified allow the network to better reproduce our experimental results. In the next step, we thus set out to validate our results through independent experimental testing of one of the newly identified interactions.

## A GSK3-mediated feedback loop in the AKT pathway

The first link added in our network reconstruction procedure connected JAK and FGFR (Fig 2B) and might mediate the crosstalk

from LIF to ERK signaling, which has been described previously in mESCs (Paling *et al*, 2004; Niwa *et al*, 2009). We therefore proceeded to validate the second link that was added, from GSK3 to IGFR (Fig 2B). This would constitute a feedback loop within the IGF/AKT pathway (Fig 4A) which, to our knowledge, has not been reported in mESCs so far. Moreover, this interaction might mediate crosstalk between the WNT and AKT pathways, since GSK3 is inhibited by WNT signaling (Stamos *et al*, 2014). Addition of this link decreased the model residuals for XX from 1,515 to 971 (36% improvement) and that for XO from 1,064 to 970 (8% improvement; Fig 2B). To validate the predicted interaction, we treated XX and XO mESCs with increasing doses of the GSK3 inhibitor, CHIR99021, and measured the phosphorylation of AKT (Ser473) after 30 min of treatment. With increasing doses of GSK3 inhibitor, phosphorylation of AKT decreased in a concentration-dependent manner in both cell lines as compared to their respective DMSO-treated controls (Fig 4B and C). This provides evidence that GSK3 indeed enhances signaling through the PI3K/AKT pathway, at a node upstream of AKT. Since AKT-dependent phosphorylation of GSK3 inhibits GSK3 activity (Cross *et al*, 1995), this new link constitutes a negative feedback loop in the PI3K/AKT pathway, where GSK3 seems to activate its own inhibitor.

GSK3 has been reported to modulate several phosphatases that inactivate AKT (Al-Khouri *et al*, 2005; Li *et al*, 2009), but also to reduce mTOR activity (Inoki *et al*, 2006), which in turn has been implicated in negative feedback inhibition of IGFR (Shah & Hunter, 2006; Hsu *et al*, 2011). To distinguish whether the predicted feedback is mediated by mTOR or not, we assessed how the mTOR target P70S6K would respond to GSK3 inhibition (Fig 4D). If the predicted feedback is mediated by mTOR, P70S6K phosphorylation should increase upon GSK3 inhibition, while an mTOR-independent feedback should result in a decrease. Phosphorylation of P70S6K clearly decreased upon GSKi treatment in XX and XO ESCs (Fig 4E and F). This implies that the feedback effect is independent of mTOR and p70S6K. In summary, we could validate a newly predicted interaction within the mESC signaling network, which might constitute feedback regulation and allow signaling crosstalk.

## Identification of X-dosage effects on the mESC signaling network

Having completed the network structure, we next aimed at identifying the links and pathways with differential activity in XX and XO mESCs. For each linear pathway, crosstalk, and feedback loop, we combined the parameters associated with all links in the path to form composite parameters, which we called pathway coefficients (see Materials and Methods for details). Through a profile-likelihood approach (Raue *et al*, 2009), we then computed 95% confidence intervals of these coefficients for each cell line (Fig 5A). We then identified those coefficients that were significantly different between the XX and the XO model, as their confidence intervals did not overlap (Figs 5B and C, and EV3A).

With this approach, we found three X chromosome dosage-sensitive pathways. Signaling through the FGF4/ERK pathway and through the ActA/SMAD2 pathway was significantly stronger in XO as compared to XX cells (Fig 5B). This suggests that XO cells respond more strongly to the differentiation triggers ActA and FGF4, potentially explaining why they differentiate more readily as compared to XX mESCs. Apart from these, the coefficient of the feedback

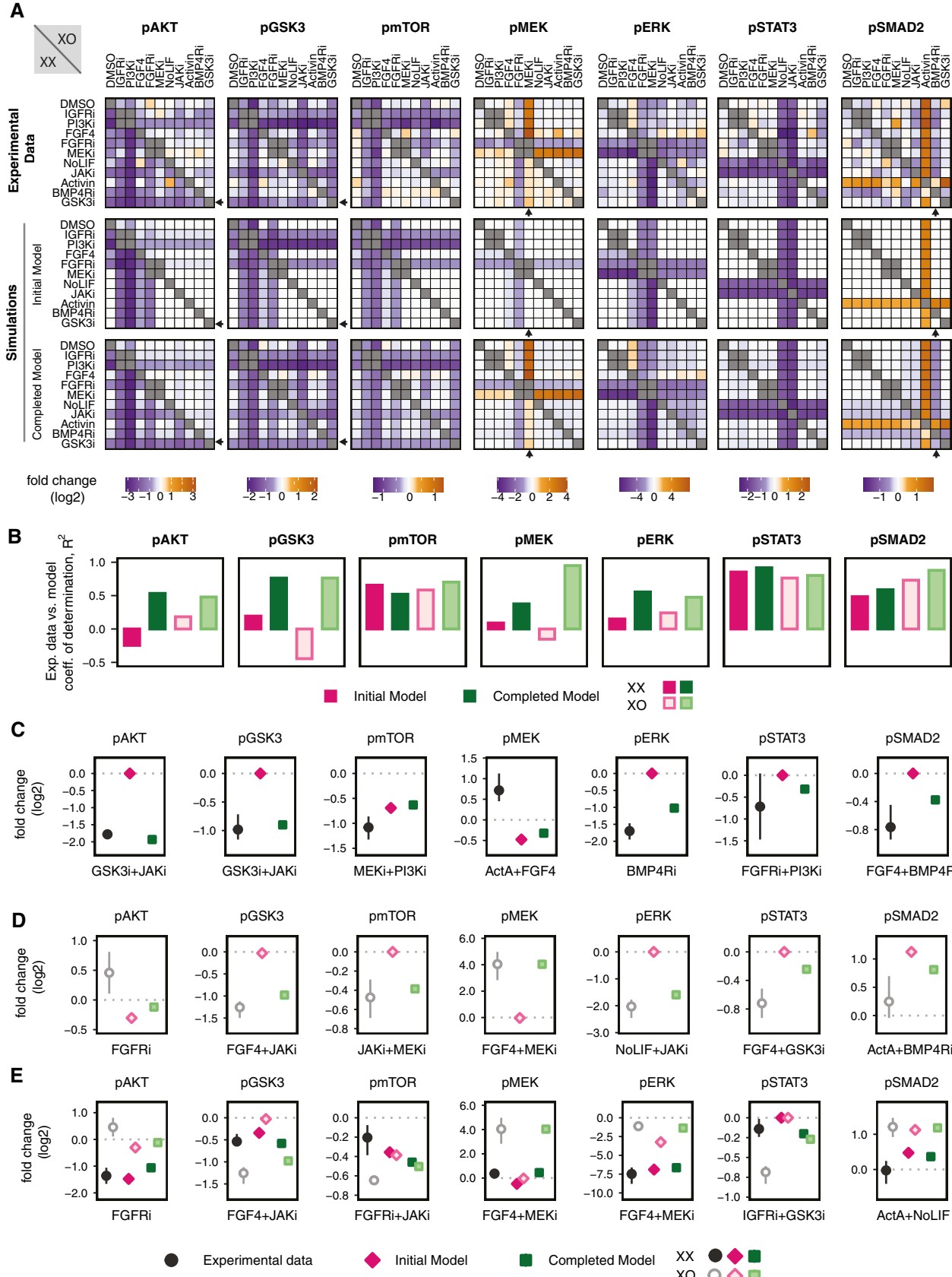

Figure 3.

**Figure 3.  The completed model can reproduce the perturbation data set.**

A       Fold change of each treatment relative to the DMSO control measured experimentally (top), simulated with the initial model (middle) and simulated with the completed model (bottom). The lower left and the upper right triangle of each heatmap corresponds to XX and XO cells, respectively. Small arrows mark example data points where model completion clearly improved the fit to the experimental data.

B       Coefficient of determination $R^2$, calculated between the experimentally determined systematic perturbation data and results from simulation of the initial or the completed models of XX and XO cells.

C, D    For each readout, the treatment with the highest discrepancy between experimental data and initial model simulations is shown for XX (C) and XO cells (D).

E       For each readout, the treatment to which the response of XX and XO cells was found to be maximally different in the experimental measurements are shown: log2 fold change in experiments and their simulated values in the initial and completed model of the two cell lines. Data points that were significantly different between XX and XO were filtered (two-sided *t*-test, $P < 0.05$), and the one with the maximum difference in the mean value, for each analyte, has been plotted. In the case of pAKT, none of the treatments passed this threshold of significance, and hence the stringency was reduced to $P < 0.1$.

Data information: For the experimental data in (C–E) mean and s.d. three independent experiments are shown.
Source data are available online for this figure.

loop in the MAPK pathway ($MEK^- > ERK^- > RAF^- > MEK$), which carried a negative sign implying an inhibitory feedback, was stronger in XO cells as evident by its higher absolute value in the XO line as compared to XX (Fig 5C). To test these model predictions, we proceeded to validate two of these identified sex differences experimentally.

**The ActA response is stronger in XO than in XX mESCs**

One key difference between XX and XO mESCs our network reconstruction approach had identified was a stronger response in XO cells to ActA stimulation (Figs 5B and 6A). Since ActA drives differentiation of mESCs (Fei *et al*, 2010), differential sensitivity to this growth factor could explain more efficient differentiation observed in mESCs with one X chromosome (Schulz *et al*, 2014). To validate this prediction experimentally, we treated XX and XO mESCs with increasing concentrations of ActA as well as an Activin receptor inhibitor (ACTRi) SB505124 and measured phosphorylation of SMAD2 (Fig 6B and C). In untreated cells, pSMAD2 levels were slightly higher in XX compared to XO mESCs. Upon ACTRi treatment, we observed a slight decrease, but SMAD2 phosphorylation remained visible, possibly stimulated by other members of the TGFb family, which also signal through SMAD2/3. Stimulation with ActA increased pSMAD2 levels stronger (~5-fold) in XO cells compared to XX mESCs (2.3-fold) at the commonly used concentration of 30 ng/ml, resulting in two-fold higher levels in XO compared to XX. We also treated male mESCs (E14) with this ActA dose and observed a strong increase in pSMAD phosphorylation (4-fold) similar to XO cells (Fig 6D and E). Cells with a single X chromosome are therefore indeed more sensitive to ActA stimulation.

To assess whether this differential sensitivity would have consequences on the cell state, we assessed the transcriptional response after 24 and 48 h of ActA treatment. Here we included two XX/XO cell line pairs (1.8, Pgk12.1) and a male line (E14) in the comparison (Figs 6F and EV4). We tested two pluripotency factors (Oct4, Nanog) and two differentiation markers (Fgf5, Otx2), but only Fgf5 seemed to respond to ActA. Both XX lines expressed very low levels of Fgf5 and did not respond to ActA. Cells with one X chromosome by contrast (XO, XY) exhibited higher basal levels, which increased further upon ActA treatment. To analyze whether the number of X chromosomes present in the cell might modulate the response to ActA treatment, we performed a two-way ANOVA. For Fgf5, we indeed found a statistically significant interaction between ActA treatment and X-chromosome number ($P = 0.02$, Fig 6F). The

pro-differentiation trigger ActA thus indeed evoked a stronger response in the cells with one X chromosome. It appears as though the response of XX cells to ActA treatment is kept under control by a differentiation checkpoint that mandates X-dosage compensation before the cells can initiate differentiation.

**Differential feedback inhibition in the MAPK pathway is mediated by inhibitory RAF phosphorylation**

Our analysis predicted that negative feedback regulation within the MAPK pathway is stronger in cells with one X chromosome compared to XX mESCs (Figs 5C and 7A), such that pMEK will respond more strongly to MEK inhibition in XO cells. To validate this prediction, we treated XX, XO, and XY cells with MEK inhibitor U0126 for 24 h (Fig 7B and C). As predicted, MEK inhibition generally led to a stronger increase in pMEK levels in XO and XY cells (2–3-fold) compared to XX mESCs (< 2-fold; Fig 7C, bottom). Comparison of absolute phosphorylation levels between cell lines however revealed that basal pMEK levels were higher in XX compared to XO and XY cells (Fig 7C, top), which we have previously attributed to partial MAPK inhibition in XX mESCs based on target gene analysis (Schulz *et al*, 2014). The stronger response of XO and XY cells to MEK inhibition thus appears to be due to endogenous inhibition of the pathway in XX cells. This would result in reduced feedback inhibition and could explain the higher basal levels in MEK phosphorylation in XX cells.

To investigate negative feedback inhibition in more detail, we analyzed a well-characterized strong feedback loop, which is mediated by ERK-dependent phosphorylation of RAF1 at inhibitory sites Ser289/296/301 (Fig 7A, inset). This renders RAF1 inactive and desensitized to further activation via RAS (Dougherty *et al*, 2005; Dhillon *et al*, 2007). When assaying phosphorylation of RAF1 at the inhibitory sites Ser289/296/301 upon MEK inhibition, pRAF1 decreased more strongly in XO and XY cells compared to XX mESCs (Fig 7D, bottom). This observation further supports our model prediction that negative feedback regulation is more active in mESCs with one X chromosome. Whether the endogenous MAPK inhibition in XX cells is also mediated by inhibitor RAF1 phosphorylation cannot be concluded from our experiment, since basal pRAF1 levels were variable even between XO and XY cells (Fig 7D, top).

To further validate the model prediction, we used another MEK inhibitor (PD0325901) and performed a 48-h time-course experiment in XX and XO mESCs (Fig 7E–G). Again, MEK phosphorylation increased significantly higher in XO (30- to 40-fold) compared to XX

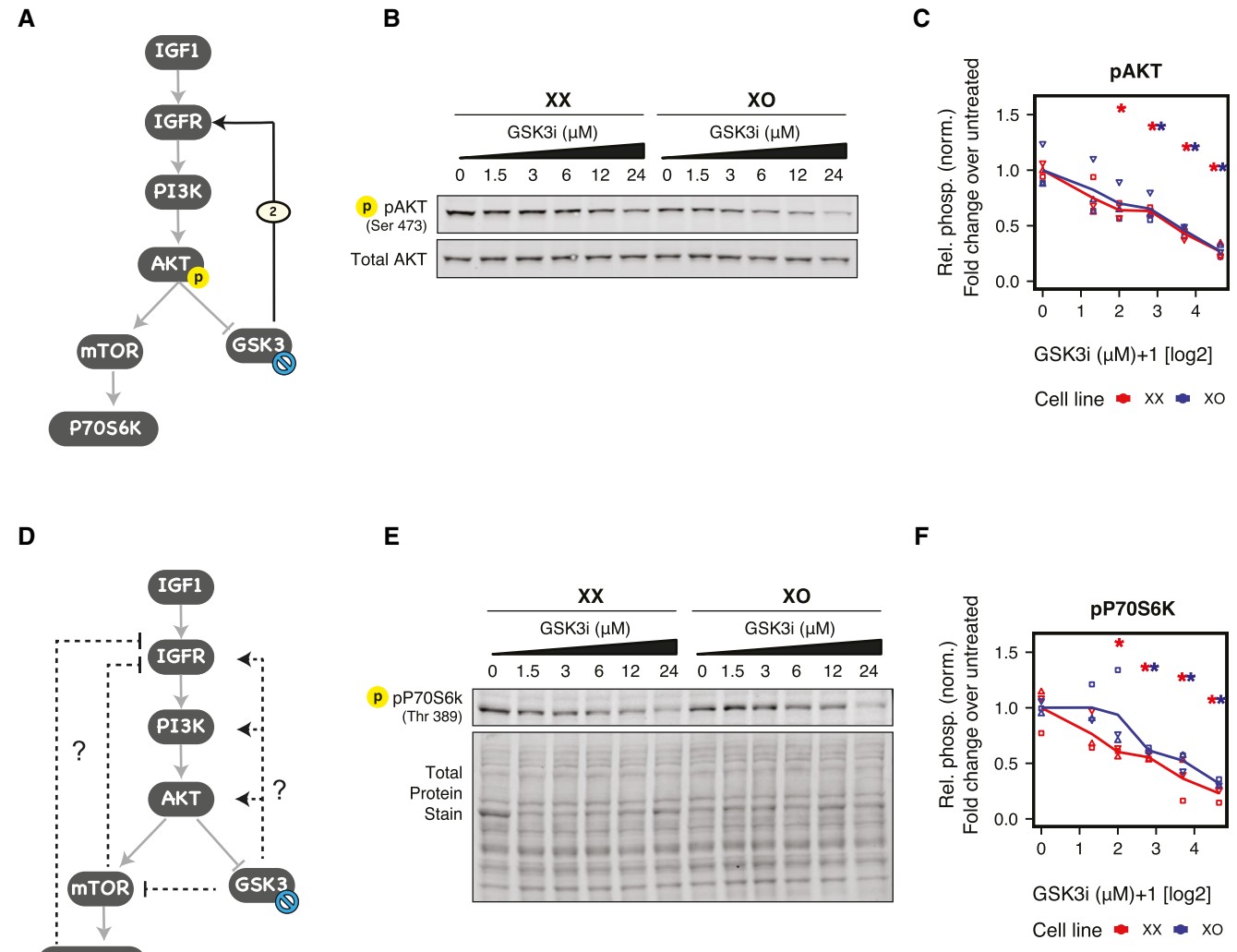

**Figure 4.  Novel link from GSK3 to IGFR, constituting a potential feedback loop in the PI3K/AKT pathway.**

A   Model extension predicts a link from GSK3 to IGFR. Prior knowledge exists for AKT-mediated phosphorylation and inactivation of GSK3. These links together suggest the existence of a negative feedback loop since GSK3 is activating its own inhibitor (AKT).

B   Representative Western blot showing the effect of increasing concentration of GSK3 inhibitor (CHIR99021) on pAKT (Ser 473) after 30 min of treatment.

C   Relative phosphorylation of AKT (pAKT/AKT) normalized over DMSO-treated control of the respective cell line.

D   Schematic representation of mTOR-dependent and mTOR-independent pathways that might mediate the predicted feedback loop between GSK3 and AKT, which can be distinguished by measuring P70S6K activity.

E   Representative Western blot showing the effect of increasing concentration of GSK3 inhibitor (CHIR99021) on pP70S6k (Thr 389).

F   Phosphorylation of P70S6K normalized to total protein stain, relative to DMSO-treated control of the respective cell line.

Data information: In (C) and (F), the mean (lines) of three independent biological replicates (symbols) are shown; asterisks indicate $P < 0.05$, unpaired $t$-test for the comparison to the DMSO-treated control.

Source data are available online for this figure.

cells (< 10-fold), associated with a stronger decrease in pRAF1 levels (Fig 7F and G, bottom). The higher effect size compared to the other MEK inhibitor (U0126) suggests that this inhibitor is more potent. Nevertheless, again XO cells showed a stronger reduction of pRAF1 upon MEKi treatment compared to XX mESCs (Fig 7F). The differential response of XX and XO cells to MEK inhibition could thus be seen using two different pharmacological inhibitors.

To test whether partial endogenous MAPK pathway inhibition in XX cells could indeed explain their altered pathway activity, we aimed to find a MEKi dose that would shift XO cells to the XX signaling state. We treated cells with different MEKi doses (PD0325901) for 24 h to mimic the constitutive MAPK inhibition in XX cells (Fig 7H and I). Full inhibition led again to similar pMEK levels in XX and XO cells. Although replicates were somewhat variable in this experiment, the same observation was made in the 24 h time point of the time course experiment (Fig 7F). Upon treatment with 4–12 nM of MEKi, pMEK levels in XO cells resembled those in untreated XX cells (Fig 7J and K). Notably, also the associated pRAF1 levels resembled those in untreated

**A**

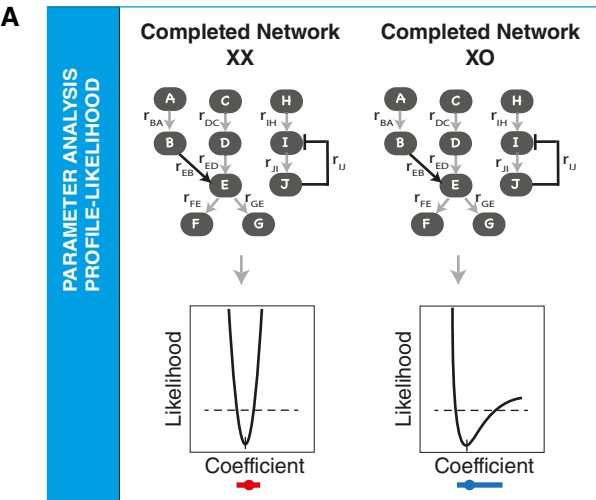

Figure 5. Identifying links with differential activity in XX and XO cells.

A    The parameters/local response coefficients associated with the completed models for the cell lines were analyzed using profile likelihood to find the 95% confidence interval (CI) associated with the parameters.

B, C   95% CI of the composite parameters derived using profile likelihood for the canonical pathways (B) and for feedback loops (C) of the starting network. Parameters with non-overlapping CI are marked with an asterisk.

XX cells, further supporting the idea of partial MAPK inhibition in XX cells. Finally, we also calculated the pMEK fold change between the partially and fully inhibited states of XO cells, which now closely resembled the fold change found for XX mESCs (Fig 7K). The stronger feedback activity in XO cells found through our MRA approach can thus potentially be explained by partial inhibition of the pathway in XX cells.

Having validated that lower MEK activity in XX cells leads to reduced negative feedback strength, we investigated how this would affect transcription of lineage markers. We again assayed Nanog, Oct4, Fgf5, and Otx2, but only observed a clear trend for Fgf5 (Fig EV5). A two-way ANOVA revealed an interaction between X chromosome number and the response to MEKi treatment for Fgf5 and Otx2. In cells with one X chromosome, MEK inhibition seemed to lead to a reduction in Fgf5 and Otx2, while XX cells, Fgf5 levels were unchanged, while Otx2 expression even appeared to increase (Fig EV5A and B). These observations support the conclusion that X chromosome number modulates the response also at the transcriptional level. Taken together, we could show that phosphorylation of MEK and RAF1 responds more strongly to MEK inhibition in cells with one X chromosome compared to XX cells, which appears to have downstream effects on the differentiation marker Fgf5. We have thus identified an X-dosage-dependent difference in the pathway state and could identify the molecular implementation of the observed feedback regulation.

**B**

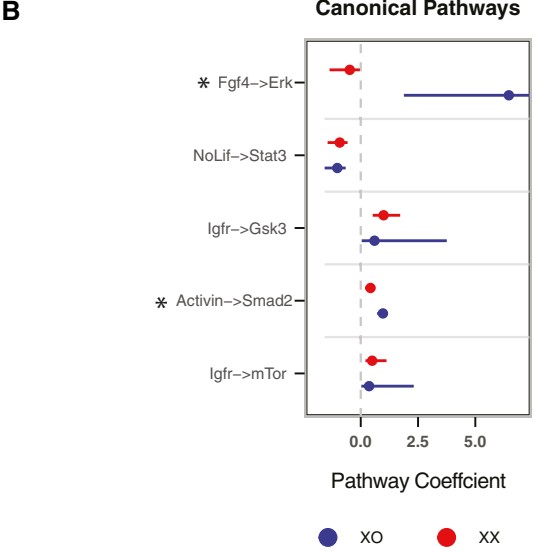

**C**

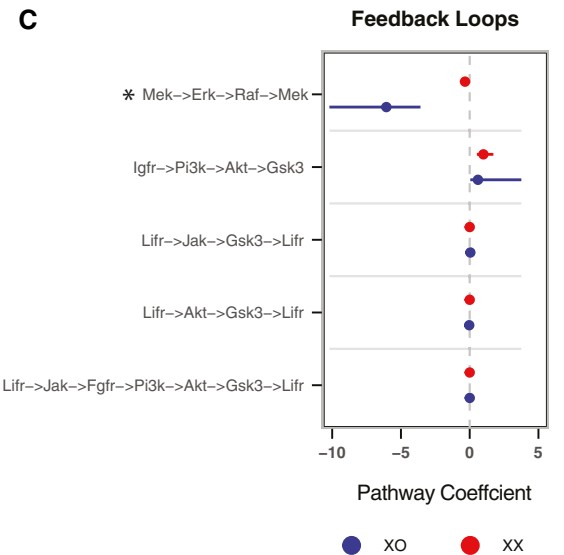

# Discussion

Here we present a comprehensive model of the signaling network in mESCs, where both network structure and model parameters were deduced from a large experimental perturbation data set. To elucidate the mechanistic basis of X-dosage dependent sex differences in mESCs, we parameterized separate models for cells with one and two X chromosomes, respectively. Our approach uncovered multiple previously described interactions that mediate crosstalk between pathways and feedback regulation, but also predicted previously unknown interactions. One of these, an activation of the AKT/mTOR pathway by GSK3, was validated experimentally. Comparison of XX and XO models revealed that cells with one X chromosome react more sensitively to the differentiation cues FGF4 and ActA. Moreover, we found differences in feedback strength within the MEK/ERK pathway, which is mediated by inhibitory RAF1 phosphorylation and which we could attribute to partial inhibition of the pathway in XX cells. Our results thus shed light on the complex interactions within the signaling network in mESCs and elucidate how X-dosage effects modulate the cells' differentiation potential.

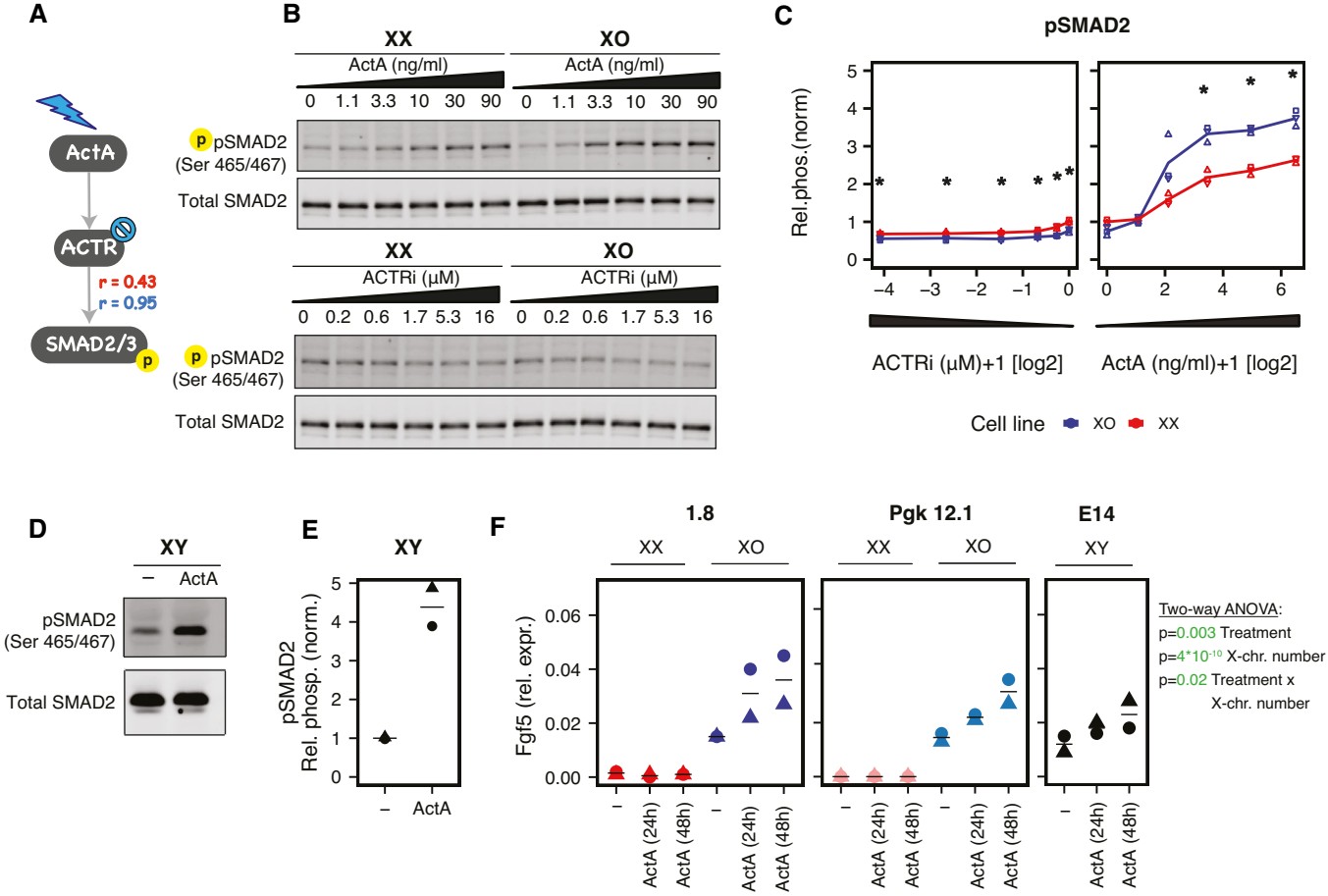

**Figure 6. Response to ActA is stronger in XO compared to XX cells.**

A    The model predicts that ActA-induced SMAD2 activation is stronger in XO (blue, r = 0.95) as compared to XX cells (red, r = 0.43).

B, C    Treatment of XX and XO mESCs with different doses of ActA and Activin receptor inhibitor (ACTRi, SB505124) for 30 min as indicated. Representative Western blot (B) and quantification (C) of SMAD2 phosphorylation at Ser465/467 normalized over total SMAD2. Asterisks indicate P < 0.05, unpaired t-test comparing XX and XO cells.

D, E    Treatment of XY mESCs (E14) with 30 ng/ml ActA for 30 min. Representative Western blot (D) and quantification (E) as in (B, C).

F    Treatment of two XX/XO cell lines pairs (1.8, Pgk12.1) and XY mESCs (E14) with 30 ng/ml ActA for 24 or 48 h as indicated. Fgf5 expression was assayed by qRT–PCR. Results of a two-way ANOVA analyzing the effects of ActA treatment and X-chromosome number on Fgf5 expression are reported, P < 0.05 are colored in green.

Data information: In (C, E, F) the mean (lines) of 3 (C) or 2 (E, F) independent biological replicates (symbols) are shown.
Source data are available online for this figure.

To our knowledge, this is the first attempt at modeling the signaling network of mESCs and investigating the impact of X-chromosome dosage on the maintenance of pluripotency and initiation of differentiation. Mathematical models to study pluripotency in stem cells have mostly focused on the involved gene regulatory networks (Herberg *et al*, 2014; Yachie-Kinoshita *et al*, 2018; Dunn *et al*, 2019). While transcriptional regulation is important to understand phenotypic outcomes based on changes in gene expression, it leaves out a critical and faster-acting layer of regulation, which is the first response of cells to environmental cues and evident in the form of changes in activation of signaling proteins.

One of the newly found interactions, which we then validated individually was activation of the AKT pathway by GSK3. This finding is in agreement with a similar observation in human cancer cells (Lu *et al*, 2011). Since GSK3 is phosphorylated by AKT, which prevents recognition for a subset of GSK3 substrates (Cross *et al*, 1995; Frame

*et al*, 2001) thereby inhibiting GSK3 activity, the newly found link could provide a negative feedback loop. Another consequence might be crosstalk between the WNT pathway, which signals via GSK3 inhibition, and the AKT/mTOR pathway. This idea is supported by our finding that mTOR-mediated phosphorylation of P70S6K is also reduced upon GSK3 inhibition. Since GSK3 inhibits WNT signaling, while activating the AKT pathway, it seems to both stabilize and destabilize the pluripotent state, since both WNT and AKT pathways block differentiation (Paling *et al*, 2004; Sato *et al*, 2004; Storm *et al*, 2009). This could potentially explain why maintenance of ES cells in the naive pluripotent state requires partial inhibition of GSK3 (Nichols & Smith, 2012; Ying & Smith, 2017). A GSK3 inhibitor is part of the so-called "2i" medium, which allows maintenance of mESCs in a homogeneous pluripotent state in chemically defined conditions (Ying *et al*, 2008). The optimal concentration of the GSK3 inhibitor CHIR99021 used in 2i culture (3 μM) only partially inhibits GSK3

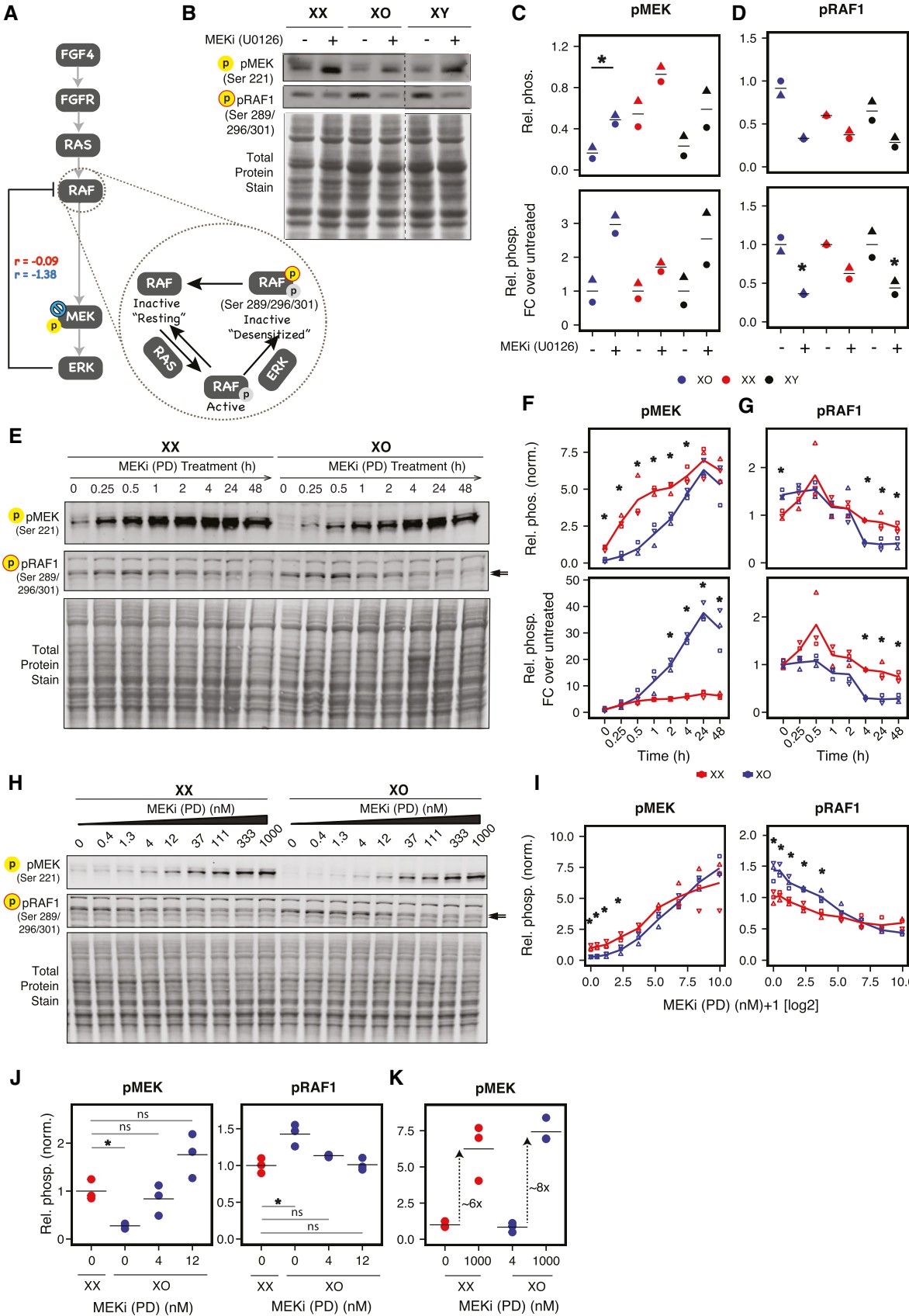

Figure 7.

**Figure 7.  Stronger RAF1-mediated feedback inhibition in the MAPK pathway in XO compared to XX cells.**

A    Model extension recovers the well-reported feedback loop in the MAPK pathway. The magnitude of the parameter associated with this feedback link is stronger in XO (blue, $r = -1.38$) as compared to XX (red, $r = -0.09$). Inset illustrates ERK-mediated hyperphosphorylation of RAF1 at Ser289/296/301, which results in inactivation as well as desensitization of RAF1. We measured this inhibitory phosphorylation of RAF1 at Ser289/296/301 catalyzed by ERK.

B    Representative Western blot showing treatment with 5 μM MEK inhibitor (U0126) for 24 h on pMEK (Ser 221) and pRAF1 (hyperphosphorylated at Ser289/296/301).

C, D  Quantification of relative phosphorylation of MEK (C) and RAF1 (D) of the MEK inhibitor response in (B). The phosphoprotein signals were normalized over total protein stain. The data points are either normalized over the maximum values of pMEK and pRAF1 respectively (top) or as fold change relative to the DMSO-treated controls for the respective cell line (bottom). Asterisks indicate $P < 0.05$ of a two-sided paired $t$-test (top) and a one-sample test for the fold change (bottom).

E    Representative Western blot showing time course (15 min to 48 h) of 1 μM MEK inhibitor (PD0325901) on pMEK (Ser 221) and pRAF1 (hyperphosphorylated at Ser289/296/301).

F, G  Quantification of relative phosphorylation of MEK (F) and RAF1 (G) in response to MEK inhibitor time course. The phosphoprotein signals were normalized over total protein stain. The data points are either shown relative to untreated XX cells (top) or as fold change relative to the DMSO-treated controls for the respective cell line (bottom).

H    Representative Western blot showing effect of increasing concentration of MEK inhibitor (PD0325901) on pMEK (Ser 221) and pRAF1 (Ser289/296/301), assayed after 24 h of treatment.

I, J  Quantification of relative phosphorylation of MEK and RAF1 in response to different MEK inhibitor concentrations. The phosphoprotein signals were normalized over total protein stain. A subset of data points is shown in (J) to demonstrate that treatment of XO cells with 4–12 nM MEKi results in similar MEK and RAF1 phosphorylation levels as in untreated XX cells.

K    pMEK fold change between the partially inhibited (4 nM MEKi-treated) and fully inhibited (1,000 nM MEKi-treated) state of XO cells is comparable to pMEK fold change found for XX mESCs (DMSO-treated and 1,000 nM MEKi-treated).

Data information: In (C, D, F, G, I, J, K), the mean (lines) of 3 (F, G, I, J, K) or 2 (C, D) independent biological replicates (symbols) are shown; asterisks indicate $P < 0.05$, unpaired $t$-test. Arrows in (E, H) indicate the band with the expected size for RAF1.
Source data are available online for this figure.

(Nichols & Smith, 2012; Ying & Smith, 2017). At these intermediate concentrations, the release of WNT inhibition might already be effective, while inhibition of the AKT/mTOR pathway, which is important for cell viability, can still be tolerated. Hence, the GSK3-to-IGF link represents an interesting aspect to be further investigated to understand its dual role in pluripotency control.

Female pluripotent cells differentiate more slowly than male cells, which we have previously shown to be caused by increased dosage of X-chromosomal genes (Schulz *et al*, 2014; Song *et al*, 2019). Our quantitative comparison of the signaling networks in XX and XO mESCs revealed that cells with one X chromosome respond more sensitively to the differentiation cues ActA and FGF4. For ActA we confirmed the predicted stronger response in XO cells through independent experiments. FGF4 and ActA are typically used to drive differentiation of mESCs to epiblast stem cells, which represent primed pluripotent cells of the postimplantation embryo (Gadue *et al*, 2006; Kunath *et al*, 2007; Fei *et al*, 2010). Our finding can thus explain the previously observed resistance of female mESCs to differentiation. This resistance is likely resolved once X-dosage compensation has occurred through X-chromosome inactivation and might be required for a coordinated progression of embryonic development.

While the mechanistic basis for the reduced sensitivity of XX cells to ActA remains to be elucidated, reduced sensitivity to FGF4 might be associated with a partial block in the MEK/ERK pathway. We have previously shown that a higher dose of DUSP9, an X-linked ERK phosphatase, and consequently reduced negative feedback activity in the pathway is responsible for increased MEK phosphorylation in XX cells (Genolet *et al*, 2021). Specifically, DUSP9 overexpression in male mESCs leads to increased MEK phosphorylation and reduced target gene expression. We show that partial pharmacological inhibition of MEK allows XO cells to assume an XX-like state. Since the pathway is already partially inhibited in XX cells, they respond less to further inhibition through a MEK inhibitor. Further downstream this leads to upregulation of the differentiation

marker Fgf5 exclusively in XO cells. Partial MAPK inhibition and reduced sensitivity to ActA is therefore likely to underlie the previously observed differentiation delay in mESCs with two X chromosomes (Schulz *et al*, 2014).

In conclusion, we leveraged computational modeling to study the cell signaling network implicated in the maintenance of pluripotency in mESCs and the impact of a double dosage of X-linked genes therein. We found evidence for GSK3-mediated feedback regulation of the PI3K/AKT/mTOR pathway in mESCs which is independent of previously described feedback loops mediated via mTOR. Moreover, we identified X-dosage-based differences in the response of mESCs to differentiation triggers. Cells with two active X chromosomes respond less to Activin and FGF4. It appears that the response of XX cells to differentiation triggers is restrained by a differentiation checkpoint that mandates X-dosage compensation before mESCs can exit pluripotency.

# Materials and Methods

### Cell culture

Female 1.8 XX mESCs carry a homozygous insertion of 7xMS2 repeats in Xist exon 7 and are a gift from the Gribnau lab; 1.8 XO mESCs are a sub-clone of this cell line that has lost an X chromosome (Schulz *et al*, 2014). The correct X chromosome number was regularly confirmed by RNA-FISH for X-linked genes.

Both mESC lines were grown on gelatin-coated flasks (Millipore, 0.1%) in serum-containing ES cell medium (DMEM (Sigma), 15% ESC-grade FBS (PAN-Biotech), 0.1 mM ß-mercaptoethanol (Sigma), 1,000 U/ml leukemia inhibitory factor (LIF, Merck)), and cultured at 37°C in a 5% $CO_2$ (v/v) incubator. Cells were passaged every 48 h, and the medium was changed every 24 h. Cells were cultured without antibiotics and routinely assessed for mycoplasma contamination by PCR.

For all perturbation and validation experiments, 1.8 XX and 1.8 XO cells were seeded in 12-well plates at a density of $4.2 \times 10^4$ and $3.4 \times 10^4$ cells/cm$^2$ respectively. Protein lysates were collected 48 h later, after treatment with growth factors or inhibitors for the indicated time periods.

## Experimental perturbations

Cells were treated with one perturbation agent, a combination of two, or DMSO at 0.3%. Biological replicates were performed in independent experiments on separate days. For the systematic perturbation experiments, all treatments were performed for 30 min. We used 10 perturbation agents, which included seven small molecule inhibitors and three growth factors (Table EV1): IGFR inhibitor OSI-906/ Linsitinib (10 μM, Selleckchem, S1091), PI3K inhibitor LY294002 (5 μM, Selleckchem, S1105), FGF4 (10 ng/ml, PeproTech), FGFR inhibitor CH5183284/Debio-1347 (0.2 μM, Selleckchem, S7665), MEK inhibitor U0126 (5 μM, R&D Systems, 5/1/1144), JAK Inhibitor I (1 μM, Calbiochem, 420099), Activin A (15 ng/ml, PeproTech), BMP4 receptor inhibitor LDN-193189 (1 μM, StemMACS, 130-106-540), and GSK3 inhibitor CHIR99021/CT99021 (6 μM, Axon, 1386). Since mESCs are cultured in a LIF-containing culture medium, the effect of LIF was studied by withdrawing LIF for a duration of 30 min (referred to as NoLIF). The 10 perturbation agents were used individually and as a combination of two perturbations, making a total of 53 treatments per cell line. Two combination treatments (IGFRi + PI3Ki and MEKi + FGFRi) were left out since both perturbations would target the same pathway (Fig 1B, inset).

Cell lysates were harvested with Bio-Plex Pro Cell signaling Reagent Kit as per the supplier's protocol. Briefly, cells were washed 1× with ice-cold PBS and lysed with cell lysis buffer (Bio-Rad) supplemented with the provided factors as per the manufacturer's protocol. The plates were shaken at 4,500 g for 30 min at 4°C. Plates were then frozen at −80°C if lysates were not collected on the same day. The lysates were transferred to 1.5 ml Eppendorf tubes and centrifuged at 4°C and 4,500 g for 20 min. The supernatant was collected in a clean tube, and protein concentration was measured using a Pierce BCA kit (Thermo Fisher Scientific). For the systematic perturbation experiment, phosphorylation of AKT, mTOR, GSK3, and MEK was quantified by using Luminex assay, while that of ERK, STAT3, and SMAD2 was quantified by immunoblotting (Fig 1B). All validation experiments were performed with immunoblotting.

## Luminex assay

For simultaneous measurement of phosphorylation of multiple proteins in lysates from perturbation experiments, we used a custom Luminex assay (Bio-Plex Pro Magnetic Cell Signaling Assays), comprising beads for phosphorylated GSK3α/β (Ser21/Ser9), MEK1 (Ser217/Ser221), mTOR (Ser2448), and AKT (Ser473). The phosphorylated residues refer to the human version of the proteins. The assay was carried out according to the manufacturer's protocol. The capture antibody beads, as well as detection antibodies and fluorescent conjugate SAPE, were diluted 1:3. The protein phosphorylation signals in the lysates were acquired with the Bio-Plex MAGPIX Multiplex Reader (Bio-Rad, Hercules, CA). The signal from the Luminex assay with these antibodies was in alignment with the response to perturbations expected based on literature.

## Immunoblotting

Protein phosphorylation was measured by immunoblotting using self-made 10% polyacrylamide gels or precast 10% Tris-Glycine Plus Midi Gels (Novex) used in the XCell4 SureLock Midi-Cell Electrophoresis System (Invitrogen), and 20–25 μg of protein was loaded per lane. Proteins were transferred to a nitrocellulose membrane (0.2 μm—Amersham, GE) using the Trans-Blot Turbo Transfer System (Bio-Rad) under semi-dry conditions at 25 V and 1.0 A for 30 min. For the validation experiments, preassembled Trans-Blot Turbo Midi 0.2 μm Nitrocellulose Transfer Packs (Bio-Rad) were used.

Membranes were blocked with Odyssey Blocking Buffer (LiCor 927-40000), diluted 1:1 in phosphate-buffered saline (PBS) for 1 h at room temperature with shaking. Thereafter the membranes were incubated overnight at 4°C with primary antibodies in Odyssey Blocking Buffer, diluted 1:1 in PBST (PBS supplemented with 0.1% Tween-20). This was followed by 4 washes with PBST and incubation with near-infrared dye-labeled secondary antibodies for 1 h at room temperature. The antibodies and their dilutions used are listed in Table EV1. After four washes with PBST and two washes with PBS, the membranes were scanned with a Li-Cor Odyssey CLx scanner. When two primary antibodies, detectable in the same channel, were used on the same membrane, prior tests were performed to assess the brightness of their signals and to rule out non-specific bands. The antibody that gave the fainter signal was incubated first, and its signal was measured. This was followed by incubation with the antibody that gave a brighter signal.

Band intensities were quantified using Image Studio Lite Version 5.2.5. The median background fluorescence was measured in the adjacent regions below and above the band. The net signal for each band was obtained by subtracting the background (given by median background signal multiplied by number of pixels in the quantification band), from the total signal of the band.

To normalize for unequal loading across lanes, total protein staining (TPS) was used. During the generation of the systematic perturbation data set, TPS was performed using Pierce Reversible Protein Stain Kit for Nitrocellulose Membranes (Thermo Fisher #24580), as per the supplier's protocol. After incubation with antibodies and scanning the signal, the membrane was washed with MilliQ water, MemCode stain was added and shaken on a platform shaker for 2 min, followed by washes with the provided destain reagent and subsequently MilliQ water. The membrane was imaged using Fusion Fx (Vilber Lourmat) and the exported tiff images were quantified using Image Studio Lite Version 5.2.5. In the validation experiments, TPS was performed using Revert 700 Total Protein Stain (Li-Cor). After the transfer of proteins from the gel, but before the blocking step, the membrane was incubated with the staining solution for 5 min, followed by two rinses with the wash solution. The membrane was then scanned in the 700 nm channel of the Li-Cor Odyssey CLx scanner. The lane intensities were quantified using the Empiria Studio Software Version 1.2.0.79 from Li-Cor.

## RNA extraction, reverse transcription, and qPCR

For gene expression profiling, cells were lysed directly in the plate by adding 1 ml of TriZol (Invitrogen). RNA was isolated using the Direct-Zol RNA Miniprep Kit (Zymo Research) following the

manufacturer's instructions with on-column DNAse treatment. 1 μg RNA was reverse transcribed using Superscript III Reverse Transcriptase (Invitrogen) and expression levels were quantified using Power SYBR Green PCR Master Mix (4368702, Thermo Fisher) normalizing to Rrm2 and Arpo. Primers used are listed in Table EV1.

### Preprocessing systematic perturbation data for modeling

#### Luminex assay: data normalization and fold change calculation
For the Bioplex data, the lxb files generated by the MAGPIX instrument were processed using the R package "lxb" to summarize the median fluorescence intensity (MFI) values for each sample (Dataset EV1, sheet 1). Since the AKT signal for XX Replicate 3 and XO Replicate 2 did not correlate with the other two replicates, they were left out from the input data for modeling. The raw MFI value of an analyte in a sample was normalized over the mean MFI value of the analyte across all samples of that replicate (Dataset EV1, sheet 2). The normalized MFI values were then used to calculate the fold change over the mean of the DMSO-treated control values within the replicate (Dataset EV1, sheet 3).

#### Immunoblotting: data normalization and fold change calculation
For the systematic perturbation data set, three analytes (pERK, pSTAT3, and pSMAD2) were measured by immunoblotting. Seven 20-well gels were needed to accommodate all the samples from the two cell lines for each biological replicate. The 53 treatment samples were distributed across the seven gels in such a manner that the same treatments from both cell lines were placed on one gel. Additionally, each gel carried DMSO-treated control samples of the two cell lines.

The band intensities from immunoblots quantified using Image Studio Lite Version 5.2.5 were normalized over the total protein stain (TPS) to account for differences in protein loading between lanes to get the normalized signal values (Dataset EV2, sheet 1). Next, this signal of each sample was divided by the mean signal across all samples on the same gel. This normalization was needed to allow comparison across gels (Dataset EV2, sheet 2). Subsequently, the fold change relative to the mean across the DMSO-treated controls of all three replicates was calculated (Dataset EV2, sheet 3).

The fold change values for the analytes measured using Luminex assay and immunoblotting combined together were used as input for network reconstruction (see below). These matrices were converted to the Minimum Information for Data Analysis in Systems Biology (MIDAS) format (described in Saez-Rodriguez et al, 2008) and capture the fold change in seven phosphoproteins upon perturbation with 53 treatments in three biological replicates for each cell line, and were used as input for modeling (Dataset EV3: XX_MIDAS.csv, Dataset EV4: XO_MIDAS.csv).

### Model construction

MRA is a method that uses systematic perturbation data to reverse-engineer the underlying regulatory network (Kholodenko et al, 1997; Bruggeman et al, 2002; Santra et al, 2018) and has been successfully applied to modeling cell signaling pathways (Klinger et al, 2013; Dorel et al, 2018, 2021; Brandt et al, 2019; Hood et al, 2019; Berlak et al, 2022). The network is described as a series of nodes connected by directed edges. Each edge has an associated local response coefficient "r," which describes how changes in the perturbed node affect the node connected by this edge, if all other links in a network are fixed. MRA estimates the local response coefficients from experimentally determined global response coefficients "R," which describe how changes in the perturbed node affect another node if all other links in the network are allowed to adapt and reach a new steady state in the presence of a persisting perturbation. The global response matrix was quantified by measuring the change in activity of measured nodes in response to a 30 min perturbation of the target node. The treatment duration was chosen such that the signaling network, which relies on posttranslational modifications, has sufficient time to reach a new steady state without extensive transcriptional changes downstream of the signaling pathways.

Network parameterization, network extension, and profile-likelihood analysis were performed using an R package STASNet, version 1.0.2 (https://github.com/molsysbio/STASNet; Dorel et al, 2018), which provides a maximum likelihood implementation of MRA as described earlier (Stelniec-Klotz et al, 2012; Klinger et al, 2013).

#### Parameterization of the starting network
As a starting network, we selected pathways known to control pluripotency, without assuming any crosstalk or feedback regulation (Fig 1). The starting network was parameterized separately using the perturbation data generated in XX and XO cells (Fig 2A), with each perturbation being described by an additional parameter (inhibitor/ligand efficiency). Local response coefficients and perturbation parameters were estimated with a maximum likelihood approach by minimizing the model residuals, given by the sum of squared differences between the predicted and measured global response coefficients scaled by the measurement error, using the Levenberg–Marquardt algorithm. Maximum likelihood implementation enables handling of incomplete perturbation data sets, which means that local response coefficients can be reliably estimated even if not every node of the network is perturbed (because of lack of specific small molecule inhibitors) or measured (because of lack of good antibodies). The local minimization was performed on $10^5$ random samples obtained by Latin hypercube sampling and the best result was chosen.

#### Network extension
Next, we optimized the network structure to further improve the fit to the data. All possible links were added individually to the network and the data was refitted as described above. For each cell type, all links that improved the fit to the data (likelihood ratio test, Benjamini–Hochberg adjusted $P < 0.005$) and originated from nodes that affect other nodes through post-translational modifications (thus excluding transcription factors and extracellular ligands) were extracted (Fig 2A). In order to keep the same network structure for both cell types, the link that resulted in most improvement of the fit for the XX and XO model was added to the network. To this end, the link that resulted in the highest relative reduction of model residuals, averaged across the XX and XO cells, was chosen. After addition of the new link, it was tested whether any other link could be removed without significant impact on model performance, using a chi-square test with degrees of freedom equal to the degrees lost by removal of the link). This was however never the case. Additional links were added until no further common links meeting the

filtering criterion could be found. To test the robustness of the results, the procedure of network extension was confirmed at different *P*-value thresholds (Fig EV2A–C). Details on the network extension procedure are provided in Table EV2.

### Parameter comparison between the XX and XO networks

To identify interactions with significantly different activity between XX and XO cells, we first identified network edges, where local response coefficients were significantly different between XX and XO cells. To this end, we performed profile likelihood calculation for the parameters of both cell line models (Raue *et al*, 2009). Briefly, the likelihood profile of each local response coefficient is generated by refitting the model with that parameter being kept constant at different values around its optimum, while relaxing all other response coefficients to find the optimal fit. A chi-square test with 8 degrees of freedom (the parameters for the seven inhibitors were not allowed to vary) was used to calculate the 95% confidence interval of the parameters. The parameters of adjacent/adjoining edges that constitute a complete path/loop were then combined through multiplication to form composite parameters or "pathway coefficients" that can be used to compare the activation level of the pathway. We calculated three kinds of such pathway coefficients: (i) coefficient for the canonical pathways (i.e. for pathways defined in the literature-derived network), where the pathway begins at the level of the extracellular ligand and ends at the most downstream protein in the pathway, (ii) coefficient for feedback loops, where multiple links together form a closed loop and (iii) coefficients for crosstalk for all continuous paths that could be formed after addition of the new links to the network between a ligand and a most downstream protein. Confidence intervals for the pathway coefficients were propagated using the parameter dependency calculated by the profile likelihood. Pathway coefficients were assumed to be significantly different if their 95% confidence intervals did not overlap.

## Data availability

All data are available as source data or in the EV Tables. All analysis scripts used are available under https://github.com/EddaSchulz/Sultana_paper.

**Expanded View** for this article is available online.

## Acknowledgements

We would like to thank Joost Gribnau for providing the 1.8 XX cell line. This work was supported by the Max Planck Research Group Leader Program, E:bio Module III—Xnet grant (Bundesministerium für Bildung und Forschung (BMBF) 031 L0072) and Human Frontier Science Program (CDA-00064/2018) to EGS; ZS was supported by the Deutsche Forschungsgemeinschaft (DFG, GRK1772, Computational Systems Biology). Open Access funding enabled and organized by Projekt DEAL.

## Author contributions

**Zeba Sultana:** Investigation; writing - original draft. **Mathurin Dorel:** Software; supervision. **Bertram Klinger:** Supervision. **Anja Sieber:** Supervision; investigation. **Ilona Dunkel:** Investigation. **Nils Blüthgen:** Supervision; funding acquisition. **Edda G Schulz:** Conceptualization; supervision; funding acquisition; writing – original draft.

## Disclosure and competing interests statement

The authors declare that they have no conflict of interest.

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
