## [Review Process File · Molecular Systems Biology]

Modelling unveils sex differences of signalling networks in mouse embryonic stem cells

Zeba Sultana, Mathurin Dorel, Bertram Klinger, Anja Sieber, Ilona Dunkel, Nils Blüthgen, and Edda Schulz

DOI: [10.15252/msb.202211510](https://doi.org/10.15252/msb.202211510)

Corresponding author(s): Edda Schulz (edda.schulz@molgen.mpg.de) , Nils Blüthgen (nils.bluethgen@charite.de)

Review Timeline:

Submission Date:	15th Dec 22
Editorial Decision:	7th Feb 23
Revision Received:	21st Jul 23
Editorial Decision:	16th Aug 23
Revision Received:	5th Sep 23
Accepted:	6th Sep 23

Editor: Maria Polychronidou

Transaction Report:

7th Feb 2023

Manuscript Number: MSB-2022-11510

Title: Modelling unveils sex differences of signalling networks in mouse embryonic stem cells

Author: Zeba Sultana

Mathurin Dorel

Klinger Bertram

Anja Sieber

Nils Blüthgen

Edda Schulz

Dear Dr. Schulz,

Thank you for submitting your work to Molecular Systems Biology. I apologize for the delay in sending you an editorial decision, which was due to the holiday season and the late arrival of the reports from the reviewers. We have now heard back from the three reviewers who agreed to evaluate your manuscript. As you will see from the reports below, the reviewers acknowledge the interest of the study. They raise however a series of concerns, which we would ask you to address in a major revision.

Without reiterating all the points raised in the reviews below, some of the more substantial issues are the following:

- The validation experiments using additional cell clones/lines, as requested by Reviewers #1 and #3.
- Reviewer #2's major point 2 regarding the validation of the findings in Figure 7 needs to be carefully addressed.
- During our pre-decision cross-commenting process (in which the reviewers are given a chance to make additional comments, including on each other's reports), both Reviewers #1 and #2 think that testing the differentiation capacity (as suggested by Reviewer# 3) would go beyond the scope of the paper and is therefore not mandatory for acceptance.

As you may already know, our editorial policy allows in principle a single round of major revision, and it is therefore essential to provide responses to the reviewers' comments that are as complete as possible. Please feel free to contact me in case you would like to discuss in further detail any of the issues raised by the reviewers.

On a more editorial level, we would ask you to address the following issues:

- Please provide a .docx formatted version of the manuscript text (including legends for main figures, EV figures and tables). Please make sure that the changes are highlighted to be clearly visible.
- Please provide individual production quality figure files as .eps, .tif, .jpg (one file per figure).
- Please provide a .docx formatted letter INCLUDING the reviewers' reports and your detailed point-by-point responses to their comments. As part of the EMBO Press transparent editorial process, the point-by-point response is part of the Review Process File (RPF), which will be published alongside your paper.
- Please note that all corresponding authors are required to supply an ORCID ID for their name upon submission of a revised manuscript.
- We replaced Supplementary Information with Expanded View (EV) Figures and Tables that are collapsible/expandable online (see examples in <http://msb.embopress.org/content/11/6/812>). A maximum of 5 EV Figures can be typeset. EV Figures should be cited as 'Figure EV1, Figure EV2' etc... in the text and their respective legends should be included in the main text after the legends of regular figures.

Additional Tables/Datasets should be labeled and referred to as Table EV1, Dataset EV1, etc. Legends have to be provided in a separate tab in case of .xls files. Alternatively, the legend can be supplied as a separate text file (README) and zipped together with the Table/Dataset file.

For the figures and tables that you do NOT wish to display as Expanded View figures, they should be bundled together with their legends in a single PDF file called *Appendix*, which should start with a short Table of Content. Each legend should be below the corresponding Figure/Table in the Appendix. Appendix figures and tables should be referred to in the main text as: "Appendix Figure S1, Appendix Figure S2, Appendix Table S1" etc. See detailed instructions regarding expanded view here: <https://www.embopress.org/page/journal/17444292/authorguide#expandedview>.

- Before submitting your revision, primary datasets (and computer code, where appropriate) produced in this study need to be deposited in an appropriate public database (see <http://msb.embopress.org/authorguide> - dataavailability

<https://www.embopress.org/page/journal/17444292/authorguide#dataavailability>).

The accession numbers and database should be listed in a formal "Data Availability" section (placed after Materials & Method) that follows the model below (see also <https://www.embopress.org/page/journal/17444292/authorguide#dataavailability>). Please note that the Data Availability Section is restricted to new primary data that are part of this study.

Data availability

-At EMBO Press we ask authors to provide source data for the main figures. Our source data coordinator will contact you to discuss which figure panels we would need source data for and will also provide you with helpful tips on how to upload and organize the files.

- Our journal encourages inclusion of *data citations in the reference list* to directly cite datasets that were re-used and obtained from public databases. Data citations in the article text are distinct from normal bibliographical citations and should directly link to the database records from which the data can be accessed. In the main text, data citations are formatted as follows: "Data ref: Smith et al, 2001". In the Reference list, data citations must be labeled with "[DATASET]". A data reference must provide the database name, accession number/identifiers and a resolvable link to the landing page from which the data can be accessed at the end of the reference. Further instructions are available at .

- We updated our journal's competing interests policy in January 2022 and request authors to consider both actual and perceived competing interests. Please review the policy <https://www.embopress.org/competing-interests> and update your competing interests if necessary.

Please use the heading "Disclosure statement and competing interests".

- All Materials and Methods need to be described in the main text. We would encourage you to use 'Structured Methods', our new Materials and Methods format. According to this format, the Material and Methods section should include a Reagents and Tools Table (listing key reagents, experimental models, software and relevant equipment and including their sources and relevant identifiers) followed by a Methods and Protocols section in which we encourage the authors to describe their methods using a step-by-step protocol format with bullet points, to facilitate the adoption of the methodologies across labs. More information on how to adhere to this format as well as downloadable templates (.doc or .xls) for the Reagents and Tools Table can be found in our author guidelines: < <https://www.embopress.org/page/journal/17444292/authorguide#researcharticleguide>>. An example of a Method paper with Structured Methods can be found here: .

-Regarding data quantification:

Please ensure to specify the name of the statistical test used to generate error bars and P values, the number (n) of independent experiments (please specify technical or biological replicates) underlying each data point and the test used to calculate p-values in each figure legend. Discussion of statistical methodology can be reported in the materials and methods section, but figure legends should contain a basic description of n, P and the test applied.

Graphs must include a description of the bars and the error bars (s.d., s.e.m.).

- Please provide a "standfirst text" summarizing the study in one or two sentences (approximately 250 characters, including space), three to four "bullet points" highlighting the main findings and a "synopsis image" (550px width and 400-600 px height, PNG format) to highlight the paper on our homepage.

Here are a couple of examples:

<https://www.embopress.org/doi/10.15252/msb.20199356>

<https://www.embopress.org/doi/10.15252/msb.20209475>

<https://www.embopress.org/doi/10.15252/msb.209495>

When you resubmit your manuscript, please download our CHECKLIST (<https://www.embopress.org/pb-assets/embosite/EMBO%20Press%20Author%20Checklist-1642513524327.xlsx>) and include the completed form in your submission.

Please note that the Author Checklist will be published alongside the paper as part of the transparent process (<https://www.embopress.org/page/journal/17444292/authorguide#transparentprocess>).

If you feel you can satisfactorily deal with these points and those listed by the referees, you may wish to submit a revised version of your manuscript. Please attach a covering letter giving details of the way in which you have handled each of the points raised by the referees. A revised manuscript will be once again subject to review and you probably understand that we can give you no guarantee at this stage that the eventual outcome will be favorable.

I look forward to receiving your revised manuscript.

Kind regards,
Jingyi

Jingyi Hou, PhD
Scientific Editor
Molecular Systems Biology

We realize that it is difficult to revise to a specific deadline. In the interest of protecting the conceptual advance provided by the work, we recommend a revision within 3 months (8th May 2023). Please discuss the revision progress ahead of this time with the editor if you require more time to complete the revisions. Use the link below to submit your revision:

IMPORTANT: When you send your revision, we will require the following items:

1. the manuscript text in LaTeX, RTF or MS Word format
2. a letter with a detailed description of the changes made in response to the referees. Please specify clearly the exact places in the text (pages and paragraphs) where each change has been made in response to each specific comment given
3. three to four 'bullet points' highlighting the main findings of your study
4. a short 'blurb' text summarizing in two sentences the study (max. 250 characters)
5. a 'thumbnail image' (550px width and max 400px height, Illustrator, PowerPoint or jpeg format), which can be used as 'visual title' for the synopsis section of your paper.
6. Please include an author contributions statement after the Acknowledgements section (see <https://www.embopress.org/page/journal/17444292/authorguide>)
7. Please complete the CHECKLIST available at (<https://bit.ly/EMBOPressAuthorChecklist>).

Please note that the Author Checklist will be published alongside the paper as part of the transparent process (<https://www.embopress.org/page/journal/17444292/authorguide#transparentprocess>).

See also figure legend guidelines: <https://www.embopress.org/page/journal/17444292/authorguide#figureformat>

9. Please note that corresponding authors are required to supply an ORCID ID for their name upon submission of a revised manuscript (EMBO Press signed a joint statement to encourage ORCID adoption). (<https://www.embopress.org/page/journal/17444292/authorguide#editorialprocess>)

Currently, our records indicate that the ORCID for your account is 0000-0003-1253-6868.

Link Not Available

The system will prompt you to fill in your funding and payment information. This will allow Wiley to send you a quote for the article processing charge (APC) in case of acceptance. This quote takes into account any reduction or fee waivers that you may be eligible for. Authors do not need to pay any fees before their manuscript is accepted and transferred to the publisher.

EMBO Press participates in many Publish and Read agreements that allow authors to publish Open Access with reduced/no publication charges. Check your eligibility: <https://authorservices.wiley.com/author-resources/Journal-Authors/open-access/affiliation-policies-payments/index.html>

*** PLEASE NOTE *** As part of the EMBO Press transparent editorial process initiative (see our Editorial at <https://dx.doi.org/10.1038/msb.2010.72>), Molecular Systems Biology publishes online a Review Process File with each accepted manuscripts. This file will be published in conjunction with your paper and will include the anonymous referee reports, your point-by-point response and all pertinent correspondence relating to the manuscript. If you do NOT want this File to be published, please inform the editorial office at msb@embo.org within 14 days upon receipt of the present letter.

Reviewer #1:

Embryonic stem cells with two X chromosomes are in a more undifferentiated naïve state that is maintained until X chromosome inactivation occurs. The paper from Sultana et al. aims at identifying signaling pathways that contribute to the different susceptibility of X0 and XX mouse embryonic stem cells (ESC) to differentiation. To this aim, Sultana et al. perform systematic perturbation of five pathways known to control pluripotency and differentiation in mouse X0 and XX ESCs. By using mathematical modelling, they identify novel connections in-between signaling pathways. Authors validate one of such novel connections, that is a GSK3-IGFR feedback loop in the AKT pathway. In addition, comparison of the signaling pathways reconstructed for X0 and XX cells highlighted a different strength of three pathways in X0 versus XX cells. Authors experimentally validated these predictions. In particular, they confirmed that the differentiation-promoting activin A pathway is stronger in X0 vs XX cells as it induces higher SMAD2 phosphorylation. They also show that the negative feedback loop MEK-ERK-RAF is stronger in X0 vs XX cells and that RAF is inhibited independently of MEK in XX cells. In summary, this paper provides insights into the mechanisms that differentially control pluripotency and differentiation in male versus female by modelling such differences with female cells having one or two X chromosomes.

Experiments are exhaustive and well-executed. It would be interesting to determine whether results obtained in X0 cells are confirmed in XY male ESCs.

Minor comments.

Page 12. The authors state " the negative feedback loop in the MAPK pathway was stronger in the X0 line as compared to XX ". The pathway coefficient is, however, lower in X0 versus XX cells. A better explanation needs to be provided.

Page 13/Figure 6B. No change in SMAD2 phosphorylation is observed upon Activin receptor (ACTR) inhibition. Can Activin A effect be mediated independently of ACTR?

Reviewer #2:

In this paper by Sultana et al., the authors perform a modeling approach of signaling networks after a perturbation screen affecting different pathways in XX and X0 ES cells. The aim is to identify commonalities and differences between the XX and X0 cells and thereby to get further insight into the impact of X-chromosome dose on signaling networks affecting pluripotency and differentiation.

The experiments in the paper are elegantly designed and provide comprehensive insight into the dependencies and crosstalks between different signaling pathways in ESCs. The authors do a very good job in describing the complexity of the pathways and their interactions in an easy to follow way by good illustrations and flow of the paper. The main weakness concern the validation experiments in Figure 7, from which several conclusions are drawn, which need to be either solidified experimentally and/or toned down in the text. Otherwise this is an excellent study, which would contribute novel insights and resources for the X-inactivation and pluripotency fields.

Major Points:

1.) In the experiments for Figure 1B/C, two different assays have been used for different signaling pathways (bead-based multiplex proteomics assay vs. two-color Western Blot) and fold changes were compared to DMSO control. As the two different assay types might have different dynamic range/sensitivity and therefore could generate different fold changes, it seems inappropriate to pool the data together in one single analysis as depicted in panel 1C. The authors should either choose only a single assay for that comparison graph, or show that similar fold-change results would be obtained for the same phosphorylation event when measured by the two different assays. Otherwise the results obtained by different methods will need to be shown in separate graphs.

2.) Fig. 7: The conclusions drawn from panel 7C/D are difficult to follow and might be partially overinterpretations of the data. For example, it is difficult to understand, if the difference in increase of pMEK between XX and X0 cells after inhibitor treatment by normalization to untreated levels (7C lower panel) is really justified and biologically relevant, as the absolute phosphorylation levels (7C upper panel) seem similar in XX and X0 cells. Validation of this experiment by other means (e.g. with the original

inhibitor used in the screen U0126) might help to increase confidence in the drawn conclusions. Also the reported increase in phosphorylation of RAF1 (panels 7D) in XX cells after 4 hours inhibitor treatment appear only to occur in 1 out of 3 replicates and therefore might be an overinterpretation, warranting validation by more replicates or at least toning down the text. Also in panel 7F, there appears to be a large spread between replicates in pMEK after MEKi treatment, making it difficult to have high confidence in the drawn conclusions. Increase in the number of replicates might help. Also it needs to be better defined, what biological replicates means in this case (experiments run on different days or side by side with different clones), as it might explain some of the variability. Panels 7G+H need statistics to be able to draw conclusions.

3.) Control for X-loss: Treatment of embryonic stem cells with MEKi has been shown to result in karyotypic abnormalities (Choi et al., Nature 2017). Therefore some of the variability of the results of female cells treated with different doses of MEKi as shown in Fig. 7 could be potentially due to spontaneous X-chromosome loss. It would be good to show by X-paint/karyotyping, if the XX cell line used remains XX in the culture conditions used for the MEKi experiments shown in this figure.

Minor Points:

1.) Page 8, Last sentence of final paragraph: For pmTOR, pSTAT3 and pSMAD2 the agreement was already good for the initial model, but was further improved in the completed model in the case of pSTAT3 and pSMAD2."

To make the sentence clearer, it should be mentioned that the agreement was not improved for pMTOR in XX cells.

2.) Page 10, last paragraph, first sentence: "The first link added in our network reconstruction procedure had already been described previously in mESCs (Paling et al, 2004; Niwa et al, 2009)."

Please mention what is the first link, so the reader does not have to guess or search for it.

3.) Page 10, last paragraph: "Addition of this link resulted in 36% improvement in the model performance for XX and 8% for XO data (Fig. 2B)."

Is this information visible in Fig. 2B or should the percentages refer to a different figure panel?

Reviewer #3:

In the present manuscript, Sultana et al. employed pharmacologic perturbation experiments and mathematical modeling approaches to characterize the effects of X-chromosome dosage on the signaling pathways known to control pluripotency and differentiation in female mouse embryonic stem cells (mESCs). The authors used modular response analysis (MRA) to infer quantitative differences between the signaling networks of mESCs carrying both (XX) or only one (XO) X chromosome, by performing growth factor stimulation/enzymatic inhibition and analyzing the phosphorylation status of key mediators of signaling pathways known to be involved in maintaining pluripotency (LIF/JAK/STAT3, IGF1/PI3K/AKT, BMP4/SMAD1/5/9 cascades) or driving differentiation (FGF4/MEK/ERK and ActivinA/SMAD2/3 cascades) of mESCs. The authors improved their signaling network model by adding known cross-talk and feedback interactions, and more notably by adding two novel putative pathway interactions (a GSK3-IGFR feedback loop in the AKT pathway and cross-talk between the Activin receptor and the RAS/MEK/ERK pathway). The authors then experimentally validated the effects of X-chromosome dosage on two distinct pathways as predicted by their model, namely a higher sensitivity to Activin A stimulation and stronger feedback regulation within the MAPK pathway in XO cells compared to XX mESCs. The importance of signaling status for pluripotency score is well documented and accepted in the field as demonstrated by the 2i condition inducing ground state pluripotency. The strategy of applying systems biology approaches to quantify and model signaling networks across different clones/mESC lines/X-activation states is an interesting idea. However, presently, the study requires major improvements.

A major concern pertains to the lack of mechanistic insights of how X-chromosome activation status informs gene dosages relevant to tuning the sensitivity to signaling perturbations. What are the genes involved? Can the observations made by the authors be explained by an epiphenomenon of differential pluripotency and differentiation capacity between two distinct clones or are the findings indeed a direct consequence of X-chromosome number? To answer this question candidate genes controlled by their endogenous promoters should be added back into the 1.8 XO cell line to confirm their mechanistic relevance in tuning signaling outputs.

The very limited focus on the 1.8 XX and 1.8 XO line only does not pay tribute to the great variability in differentiation competence between various mouse embryonic stem cell lines (mESCs) and subclones that is a well-known fact in the field. The authors should test their models systematically on multiple mESC lines representing the two different X-chromosome activation states (also including male and female mESC lines). I understand that transcriptional feedbacks might influence signaling status, however, at present one cannot exclude clone-to-clone variability independently of X-chromosome activation status as a confounding factor. It is therefore, important to ascertain the robustness of the presented findings across a variety of lines/subclones. At present, the limited validation scope limits generality of the findings and poses the risk that the conclusions of this investigation are based on a purely artefactual peculiarity of these specific cell lines/subclones. Finally, it would be important to investigate how the catalogued changes in signaling network activation impact differentiation capacity towards distinct mESC differentiation fates. Can different sensitivities be linked to biases in fate acquisition?

The data shown is clear and the experiments are reasonably well motivated, however a major limitation of the study in its current

state is that focuses exclusively on phosphor-protein levels while nonetheless claiming to derive insights into how pluripotency is regulated in mESCs. Thus, before the study is acceptable for publication the authors need to address how their findings affect mESCs pluripotency and differentiation outcomes in biologically meaningful ways. Most of the perturbation experiments performed by the authors span very short timespans, and while sensible for the purpose of detecting immediate biochemical readouts within signaling pathways are removed from allowing to detect gene expression changes typical of ESCs exiting naïve pluripotency. Even so, the experiments shown in Figure 7 do take place over 48h, in which changes in expression of e.g. Oct4, Nanog, Fgf5 and Otx2 should be well detectable. This is important in particular for their newly identified X-dosage dependent signaling network effects, i.e. Activin A sensitivity and MAPK feedback regulation.

Answers to reviewers' comments

Reviewer #1

Summary : Embryonic stem cells with two X chromosomes are in a more undifferentiated naïve state that is maintained until X chromosome inactivation occurs. The paper from Sultana et al. aims at identifying signaling pathways that contribute to the different susceptibility of X0 and XX mouse embryonic stem cells (ESC) to differentiation. To this aim, Sultana et al. perform systematic perturbation of five pathways known to control pluripotency and differentiation in mouse X0 and XX ESCs. By using mathematical modelling, they identify novel connections in-between signaling pathways. Authors validate one of such novel connections, that is a GSK3-IGFR feedback loop in the AKT pathway. In addition, comparison of the signaling pathways reconstructed for X0 and XX cells highlighted a different strength of three pathways in X0 versus XX cells. Authors experimentally validated these predictions. In particular, they confirmed that the differentiation-promoting activin A pathway is stronger in X0 vs XX cells as it induces higher SMAD2 phosphorylation. They also show that the negative feedback loop MEK-ERK-RAF is stronger in X0 vs XX cells and that RAF is inhibited independently of MEK in XX cells. In summary, this paper provides insights into the mechanisms that differentially control pluripotency and differentiation in male versus female by modelling such differences with female cells having one or two X chromosomes.

Major points :

1. Experiments are exhaustive and well-executed. It would be interesting to determine whether results obtained in X0 cells are confirmed in XY male ESCs.

We have tested several model predictions also in male mESCs in the revised version of the manuscript. Specifically, we have performed ActA treatment (Fig. 6D-F) and MEK inhibition (Fig. 7B-D, EV5) in the male mESC line E14. Overall, we observed similar behavior as in X0 cells, including a strong response to ActA and MEK inhibition.

Minor points :

2. Page 12. The authors state " the negative feedback loop in the MAPK pathway was stronger in the X0 line as compared to XX ". The pathway coefficient is, however, lower in X0 versus XX cells. A better explanation needs to be provided.

We have reformulated the sentence as follows (p.13): "... the coefficient of the feedback loop in the MAPK pathway (MEK -> ERK -> RAF -> MEK), which carried a negative sign implying an inhibitory feedback, was stronger in X0 cells as evident by its higher absolute value in the X0 line as compared to XX (Fig. 5C)." We hope that this adjustment will increase clarity.

3. Page 13/Figure 6B. No change in SMAD2 phosphorylation is observed upon Activin receptor (ACTR) inhibition. Can Activin A effect be mediated independently of ACTR?

Even high doses of the ACTRi did indeed not completely abolish SMAD2 phosphorylation. Most likely some basal stimulation of pSMAD2 is maintained by other members of the TGF β family, which signal through different receptors. This point is discussed in p.15 of the revised manuscript.

Reviewer #2

Summary : In this paper by Sultana et al., the authors perform a modeling approach of signaling networks after a perturbation screen affecting different pathways in XX and XO ES cells. The aim is to identify commonalities and differences between the XX and XO cells and thereby to get further insight into the impact of X-chromosome dose on signaling networks affecting pluripotency and differentiation.

The experiments in the paper are elegantly designed and provide comprehensive insight into the dependencies and crosstalks between different signaling pathways in ESCs. The authors do a very good job in describing the complexity of the pathways and their interactions in an easy to follow way by good illustrations and flow of the paper.

The main weakness concern the validation experiments in Figure 7, from which several conclusions are drawn, which need to be either solidified experimentally and/or toned down in the text. Otherwise this is an excellent study, which would contribute novel insights and resources for the X-inactivation and pluripotency fields.

Major points :

1. In the experiments for Figure 1B/C, two different assays have been used for different signaling pathways (bead-based multiplex proteomics assay vs. two-color Western Blot) and fold changes were compared to DMSO control. As the two different assay types might have different dynamic range/sensitivity and therefore could generate different fold changes, it seems inappropriate to pool the data together in one single analysis as depicted in panel 1C. The authors should either choose only a single assay for that comparison graph, or show that similar fold-change results would be obtained for the same phosphorylation event when measured by the two different assays. Otherwise the results obtained by different methods will need to be shown in separate graphs.

We agree that sensitivity and dynamic range might differ between the two assays used (and even within the same assay type), since both properties will also depend on the quality of the antibody. For our analysis it is mainly important that data points are quantitatively comparable for the same phospho-site across treatments, such that the main conclusions should still be valid in the presence of systematic differences between assays. Nevertheless, we have modified Fig. 1C to clearly show that two different assays were used to collect the data.

2. Fig. 7: The conclusions drawn from panel 7C/D are difficult to follow and might be partially overinterpretations of the data.

We apologize that the Figure 7 and the associated text were difficult to follow. We have performed additional experiments and changed the text to hopefully improve clarity and increase rigor (outlined in detail below).

(i) For example, it is difficult to understand, if the difference in increase of pMEK between XX and XO cells after inhibitor treatment by normalization to untreated levels (7C lower panel) is really justified and biologically relevant, as the absolute phosphorylation levels (7C upper panel) seem similar in XX and XO cells.

As correctly pointed out by the reviewer, pMEK levels become similar between XX and XO cells upon full MEK inhibition. Cells with one X chromosome respond more strongly to MEK inhibitor treatment, because the MAPK pathway is already partially inhibited through endogenous mechanisms in the basal state in XX cells. We have tried to explain this more clearly in the revised version of the manuscript. We have also added additional data suggesting that this difference is indeed biologically relevant, since MEK inhibition in XO cells appears to lead to reduced expression of the differentiation marker Fgf5, which is unaffected in XX cells (Fig. EV5).

(ii) Validation of this experiment by other means (e.g. with the original inhibitor used in the screen U0126) might help to increase confidence in the drawn conclusions.

As suggested we have performed an additional experiment using the MEK inhibitor U0126, which is shown in Fig. 7B-D in the revised version of the manuscript. Although this inhibitor seems to be less potent at the dose used, the experiment confirms the model prediction that cells with one X chromosome respond more strongly to MEK inhibitor treatment than XX mESCs, both with respect to MEK and RAF1 phosphorylation.

(iii) Also the reported increase in phosphorylation of RAF1 (panels 7D) in XX cells after 4 hours inhibitor treatment appear only to occur in 1 out of 3 replicates and therefore might be an overinterpretation, warranting validation by more replicates or at least toning down the text.

All time points (including the 4 h time point) have 3 replicates, but their values were very similar and overlapped in the plot.

(iv) Also in panel 7F, there appears to be a large spread between replicates in pMEK after MEKi treatment, making it difficult to have high confidence in the drawn conclusions. Increase in the number of replicates might help.

As correctly pointed out by the reviewer, in this experiment (now shown in Fig. 7I of the revised manuscript) one replicate (triangle, point down) generally showed a weaker response to MEK inhibition, making it difficult to conclude from that experiment, whether pMEK levels are indeed similar in the two cell lines upon strong inhibition. However, the time course experiment with the

same inhibitor (Fig. 7F) also shows similar pMEK levels after 24h of treatment, making us confident that the results are valid. This is now being discussed in the text on p. 17, last paragraph.

(v) Also it needs to be better defined, what biological replicates means in this case (experiments run on different days or side by side with different clones), as it might explain some of the variability.

Biological replicates were performed as independent experiments on different days. This is now stated in the methods section on p.22.

(vi) Panels 7G+H need statistics to be able to draw conclusions.

We have performed a statistical analysis of the experiment, which is shown in Fig. 7J-K in the revised version of the manuscript.

3. Control for X-loss: Treatment of embryonic stem cells with MEKi has been shown to result in karyotypic abnormalities (Choi et al., Nature 2017). Therefore some of the variability of the results of female cells treated with different doses of MEKi as shown in Fig. 7 could be potentially due to spontaneous X-chromosome loss. It would be good to show by X-paint/karyotyping, if the XX cell line used remains XX in the culture conditions used for the MEKi experiments shown in this figure.

Karyotype instability is indeed frequently observed in mESCs, which might be increased upon MEKi treatment. We regularly test our cell lines for the correct X-chromosome number before each experiment, which is now explicitly stated in the methods section (p. 22). Since X-chromosome loss normally progresses slowly over many passages (see Fig. 3e in Choi et al), we believe that it is unlikely to occur during a 1-2 day treatment with MEK inhibitor.

Minor points :

4. Page 8, Last sentence of final paragraph: For pmTOR, pSTAT3 and pSMAD2 the agreement was already good for the initial model, but was further improved in the completed model in the case of pSTAT3 and pSMAD2."

To make the sentence clearer, it should be mentioned that the agreement was not improved for pMTOR in XX cells.

We have modified the sentence as suggested.

5. Page 10, last paragraph, first sentence: "The first link added in our network reconstruction procedure had already been described previously in mESCs (Paling et al, 2004; Niwa et al, 2009)."

Please mention what is the first link, so the reader does not have to guess or search for it.

The sentence has been modified as follows (p.11): "The first link added in our network reconstruction procedure connected JAK and FGFR (Fig 2B) and might mediate the crosstalk from LIF to ERK signaling, which has been described previously in mESCs."

6. Page 10, last paragraph: "Addition of this link resulted in 36% improvement in the model performance for XX and 8% for XO data (Fig. 2B)."

Is this information visible in Fig. 2B or should the percentages refer to a different figure panel?

The percentages were calculated from the values plotted in Fig. 2B. To make this more clear the sentence has been modified as follows (p.11): "Addition of this link decreased the model residuals for XX from 1515 to 971 (36% improvement) and that for XO from 1064 to 970 (8% improvement) (Fig. 2B)".

Reviewer #3

Summary :

In the present manuscript, Sultana et al. employed pharmacologic perturbation experiments and mathematical modeling approaches to characterize the effects of X-chromosome dosage on the signaling pathways known to control pluripotency and differentiation in female mouse embryonic stem cells (mESCs).

The authors used modular response analysis (MRA) to infer quantitative differences between the signaling networks of mESCs carrying both (XX) or only one (XO) X chromosome, by performing growth factor stimulation/enzymatic inhibition and analyzing the phosphorylation status of key mediators of signaling pathways known to be involved in maintaining pluripotency (LIF/JAK/STAT3, IGF1/PI3K/AKT, BMP4/SMAD1/5/9 cascades) or driving differentiation (FGF4/MEK/ERK and ActivinA/SMAD2/3 cascades) of mESCs. The authors improved their signaling network model by adding known cross-talk and feedback interactions, and more notably by adding two novel putative pathway interactions (a GSK3-IGFR feedback loop in the AKT pathway and cross-talk between the Activin receptor and the RAS/MEK/ERK pathway).

The authors then experimentally validated the effects of X-chromosome dosage on two distinct pathways as predicted by their model, namely a higher sensitivity to Activin A stimulation and stronger feedback regulation within the MAPK pathway in XO cells compared to XX mESCs. The importance of signaling status for pluripotency score is well documented and accepted in the field as demonstrated by the 2i condition inducing ground state pluripotency.

The strategy of applying systems biology approaches to quantify and model signaling networks across different clones/mESC lines/X-activation states is an interesting idea. However, presently, the study requires major improvements.

Major points :

1. A major concern pertains to the lack of mechanistic insights of how X-chromosome activation status informs gene dosages relevant to tuning the sensitivity to signaling perturbations. What are the genes involved? Can the observations made by the authors be explained by an epiphenomenon of differential pluripotency and differentiation capacity between two distinct clones or are the findings indeed a direct consequence of X-chromosome number? To answer this question candidate genes controlled by their endogenous promoters should be added back into the 1.8 XO cell line to confirm their mechanistic relevance in tuning signaling outputs.

We agree that it is an interesting question how the presence of two X chromosomes leads to differences in the cellular signaling network. While we believe that this question goes beyond the scope of the current study, we have addressed it in a previous publication (Genolet et al, Gen Biol, 2021), where we could show that over-expression of the X-linked factor *Dusp9* leads to increased MEK phosphorylation in male mESCs. We discuss this now in more detail in the discussion (p. 21).

2. The very limited focus on the 1.8 XX and 1.8 XO line only does not pay tribute to the great variability in differentiation competence between various mouse embryonic stem cell lines (mESCs) and subclones that is a well-known fact in the field. The authors should test their models systematically on multiple mESC lines representing the two different X-chromosome activation states (also including male and female mESC lines).

I understand that transcriptional feedbacks might influence signaling status, however, at present one cannot exclude clone-to-clone variability independently of X-chromosome activation status as a confounding factor. It is therefore, important to ascertain the robustness of the presented findings across a variety of lines/subclones. At present, the limited validation scope limits generality of the findings and poses the risk that the conclusions of this investigation are based on a purely artefactual peculiarity of these specific cell lines/subclones.

We have performed additional experiments, both including a male mESC line (E14) as well as another XX/XO pair (Pgk12.1). We could confirm that male mESCs respond to ActA treatment similarly to XO cells (~4-fold increase in pSMAD2, new Fig. 6D-E vs B-C). Moreover, male mESC also respond more strongly to MEK inhibition than XX cells (new Fig. 7B-D). Moreover, in the analysis of transcriptional consequences (see answer to comment 4) we included two XX/XO cell line pairs and the XY line. Here we could show that all 3 cell lines with one X chromosome tend to show an increase in *Fgf5* expression upon ActA treatment and a reduction in response to MEK inhibition.

3. Finally, it would be important to investigate how the catalogued changes in signaling network activation impact differentiation capacity towards distinct mESC differentiation fates. Can different sensitivities be linked to biases in fate acquisition?

It would indeed be an interesting follow-up experiment to assess the differentiation competence in different lineages. However, this would go beyond the scope of the current manuscript and should be investigated in a future study.

4. The data shown is clear and the experiments are reasonably well motivated, however a major limitation of the study in its current state is that it focuses exclusively on phosphor-protein levels while nonetheless claiming to derive insights into how pluripotency is regulated in mESCs. Thus, before the study is acceptable for publication the authors need to address how their findings affect mESCs pluripotency and differentiation outcomes in biologically meaningful ways. Most of the perturbation experiments performed by the authors span very short timespans, and while sensible for the purpose of detecting immediate biochemical readouts within signaling pathways are removed from allowing to detect gene expression changes typical of ESCs exiting naïve pluripotency. Even so, the experiments shown in Figure 7 do take place over 48h, in which changes in expression of e.g. Oct4, Nanog, Fgf5 and Otx2 should be well detectable. This is important in particular for their newly identified X-dosage-dependent signaling network effects, i.e. Activin A sensitivity and MAPK feedback regulation.

We were happy to follow this suggestion and have performed additional experiments to address this question. As suggested we have assayed the pluripotency factors Nanog and Oct4 as well as the differentiation markers Fgf5 and Otx2 in response to ActA and MEK inhibitor treatment. We can show that cells with one X chromosome upregulate Fgf5 in response to ActA and reduce Fgf5 in response to MEKi treatment, while XX cells do not respond.

16th Aug 2023

Manuscript Number: MSB-2022-11510R

Title: Modelling unveils sex differences of signalling networks in mouse embryonic stem cells

Dear Dr. Schulz,

Thank you for sending us your revised manuscript. My colleague Jingyi Hou is on maternity leave and I took over handling your manuscript. We have now heard back from the three reviewers who were asked to evaluate your revised study. As you will see below, the reviewers are satisfied with the performed revisions and support publication. Some very minor issues are still listed by the reviewers and we would ask you to address them alongside some remaining editorial issues listed below. We can then formally accept the manuscript for publication.

- Our data editors have noted some missing information in the figure legends, please see the attached .doc file. Please make all requested text changes using the attached file and *keeping the "track changes" mode* so that we can easily access the edits made.
- Please provide 5 keywords.
- There are callouts to "Supplementary Tables S2 and S3", please correct to the corresponding EV Tables of Datasets.
- Tables EV2-EV5 are rather complex and should be renamed Dataset EV1-EV4. Table EV6 should be renamed as Table EV2. For all Tables and Datasets: please add the description of each table/dataset in the EV Table/Dataset file itself i.e. in a separate sheet for Excel files or as a README.txt file zipped together with the Dataset for csv files.
- Our data integrity analyst noticed some irregularities in Figure EV1A (see attached files). Perhaps this is due to some issue during image acquisition? We would ask you to replace this panel by a better quality image that does not show these irregularities.
- The synopsis image is rather complex and not all labels display well at the final size required. Please resupply the image in a .jpg or .png format and make sure that all labels are readable. It needs to be exactly 550 px wide, and the height ideally < 500 px.

Please resubmit your revised manuscript online, with a covering letter listing amendments and responses to each point raised by the referees. Please resubmit the paper **within one month** and ideally as soon as possible. If we do not receive the revised manuscript within this time period, the file might be closed and any subsequent resubmission would be treated as a new manuscript. Please use the Manuscript Number (above) in all correspondence.

Click on the link below to submit your revised paper.

Kind regards,

Maria

Maria Polychronidou, PhD
Senior Editor
Molecular Systems Biology

If you do choose to resubmit, please click on the link below to submit the revision online before 15th Sep 2023.

IMPORTANT: When you send your revision, we will require the following items:

1. the manuscript text in LaTeX, RTF or MS Word format
2. a letter with a detailed description of the changes made in response to the referees. Please specify clearly the exact places in the text (pages and paragraphs) where each change has been made in response to each specific comment given
3. three to four 'bullet points' highlighting the main findings of your study
4. a short 'blurb' text summarizing in two sentences the study (max. 250 characters)
5. a 'thumbnail image' (550px width and max 400px height, Illustrator, PowerPoint or jpeg format), which can be used as 'visual title' for the synopsis section of your paper.
6. Please include an author contributions statement after the Acknowledgements section (see <https://www.embopress.org/page/journal/17444292/authorguide#manuscriptpreparation>)
7. Please complete the CHECKLIST available at (<https://bit.ly/EMBOPressAuthorChecklist>). Please note that the Author Checklist will be published alongside the paper as part of the transparent process (<https://www.embopress.org/page/journal/17444292/authorguide#transparentprocess>).
8. When assembling figures, please refer to our figure preparation guideline in order to ensure proper formatting and readability in print as well as on screen:
<https://bit.ly/EMBOPressFigurePreparationGuideline>
See also figure legend guidelines: <https://www.embopress.org/page/journal/17444292/authorguide#figureformat>
9. Please note that corresponding authors are required to supply an ORCID ID for their name upon submission of a revised manuscript (EMBO Press signed a joint statement to encourage ORCID adoption). (<https://www.embopress.org/page/journal/17444292/authorguide#editorialprocess>)
Currently, our records indicate that the ORCID for your account is 0000-0003-1253-6868.

Link Not Available

The system will prompt you to fill in your funding and payment information. This will allow Wiley to send you a quote for the article processing charge (APC) in case of acceptance. This quote takes into account any reduction or fee waivers that you may be eligible for. Authors do not need to pay any fees before their manuscript is accepted and transferred to the publisher.

*** PLEASE NOTE *** As part of the EMBO Press transparent editorial process initiative (see our Editorial at <https://dx.doi.org/10.1038/msb.2010.72> , Molecular Systems Biology will publish online a Review Process File to accompany accepted manuscripts. When preparing your letter of response, please be aware that in the event of acceptance, your cover letter/point-by-point document will be included as part of this File, which will be available to the scientific community. More information about this initiative is available in our Instructions to Authors. If you have any questions about this initiative, please contact the editorial office (msb@embo.org).

Reviewer #1:

The authors have properly addressed my comments and extended their validation to male XY mESC and further XX/X0 lines showing robustness of their findings.

Reviewer #2:

The authors have made an effort to respond to the reviewers questions either experimentally or by rewriting / explaining the findings better. In particular, addition of male and other XX/XO cell lines and using a second MEKi for validation provided more robustness in the drawn interpretations. I therefore believe that the paper is now sufficiently improved to address most of the reviewer's concerns.

Minor comment: In figure 7F there is an additional "F" label floating between the upper and lower panel, which should be removed.

Reviewer #3:

Great work by the authors. I feel confident recommending the revised version of this manuscript for publication.

All editorial and formatting issues were resolved by the authors.

6th Sep 2023

RE: MSB-2022-11510RR, Modelling unveils sex differences of signalling networks in mouse embryonic stem cells

Dear Edda,

Thank you again for sending us your revised manuscript. We are now satisfied with the modifications made and I am pleased to inform you that your paper has been accepted for publication.

*** PLEASE NOTE *** As part of the EMBO Publications transparent editorial process initiative (see our Editorial at <https://dx.doi.org/10.1038/msb.2010.72>), Molecular Systems Biology publishes online a Review Process File with each accepted manuscripts. This file will be published in conjunction with your paper and will include the anonymous referee reports, your point- by-point response and all pertinent correspondence relating to the manuscript. If you do NOT want this File to be published, please inform the editorial office at msb@embo.org within 14 days upon receipt of the present letter.

Should you be planning a Press Release on your article, please get in contact with msb@wiley.com as early as possible, in order to coordinate publication and release dates.

LICENSE AND PAYMENT:

All articles published in Molecular Systems Biology are fully open access: immediately and freely available to read, download and share.

Molecular Systems Biology charges an article processing charge (APC) to cover the publication costs. You, as the corresponding author for this manuscript, should have already received a quote with the article processing fee separately. Please let us know in case this quote has not been received.

Once your article is at Wiley for editorial production you will receive an email from Wiley's Author Services system, which will ask you to log in and will present you with the publication license form for completion. Within the same system the publication fee can be paid by credit card, an invoice or pro forma can be requested.

Payment of the publication charge and the signed Open Access Agreement form must be received before the article can be published online.

Molecular Systems Biology articles are published under the Creative Commons licence CC BY, which facilitates the sharing of scientific information by reducing legal barriers, while mandating attribution of the source in accordance to standard scholarly practice.

Proofs will be forwarded to you within the next 2-3 weeks.

Thank you very much for submitting your work to Molecular Systems Biology.

Kind regards,

Maria

Maria Polychronidou, PhD
Senior Editor
Molecular Systems Biology